# Dietary probiotic and synbiotic supplementation starting from maternal gestation improves muscular lipid metabolism in offspring piglets by reshaping colonic microbiota and metabolites

Qian Zhu,[1,2] Md. Abul Kalam Azad,[1] Ruixuan Li,[1,2] Chenjian Li,[1] Yang Liu,[1] Yulong Yin,[1,2] Xiangfeng Kong[1,2]

**ABSTRACT**    Probiotics and synbiotics have been intensively used in animal husbandry due to their advantageous roles in animals' health. However, there is a paucity of research on probiotic and synbiotic supplementation from maternal gestation to the postnatal growing phases of offspring piglets. Thus, we assessed the effects of dietary supplementation of these two additives to sows and offspring piglets on skeletal muscle and body metabolism, colonic microbiota composition, and metabolite profiles of offspring piglets. Pregnant Bama mini-pigs and their offspring piglets (after weaning) were fed either a basal diet or a basal diet supplemented with antibiotics, probiotics, or synbiotics. At 65, 95, and 125 days old, eight pigs per group were euthanized and sampled for analyses. Probiotics increased the intramuscular fat content in the psoas major muscle (PMM) at 95 days old, polyunsaturated fatty acid (PUFA) and n-3 PUFA levels in the longissimus dorsi muscle (LDM) at 65 days old, C16:1 level in the LDM at 125 days old, and upregulated *ATGL*, *CPT-1*, and *HSL* expressions in the PMM at 65 days old. Synbiotics increased the plasma HDL-C level at 65 days old and TC level at 65 and 125 days old and upregulated the *CPT-1* expression in the PMM at 125 days old. In addition, probiotics and synbiotics increased the plasma levels of HDL-C at 65 days old, CHE at 95 days old, and LDL-C at 125 days old, while decreasing the C18:1n9t level in the PMM at 65 days old and the plasma levels of GLU, LDH, and TG at 95 days old. Microbiome analysis showed that probiotic and synbiotic supplementation increased colonic Actinobacteria, Firmicutes, Verrucomicrobia, *Faecalibacterium*, *Pseudobutyrivibrio*, and *Turicibacter* abundances. However, antibiotic supplementation decreased colonic Actinobacteria, Bacteroidetes, *Prevotella*, and *Unclassified_Lachnospiraceae* abundances. Furthermore, probiotic and synbiotic supplementation was associated with alterations in 8, 7, and 10 differential metabolites at three different age stages. Both microbiome and metabolome analyses showed that the differential metabolic pathways were associated with carbohydrate, amino acid, and lipid metabolism. However, antibiotic supplementation increased the C18:1n9t level in the PMM at 65 days old and xenobiotic biodegradation and metabolism at 125 days old. In conclusion, sow-offspring's diets supplemented with these two additives showed conducive effects on meat flavor, nutritional composition of skeletal muscles, and body metabolism, which may be associated with the reshaping of colonic microbiota and metabolites. However, antibiotic supplementation has negative effects on colonic microbiota composition and fatty acid composition in the PMM.

**IMPORTANCE**    The integral sow-offspring probiotic and synbiotic supplementation improves the meat flavor and the fatty acid composition of the LDM to some extent. Sow-offspring probiotic and synbiotic supplementation increases the colonic

Address correspondence to Xiangfeng Kong, nnkxf@isa.ac.cn.

The authors declare no conflict of interest.

See the funding table on p. 30.

beneficial bacteria (including Firmicutes, Verrucomicrobia, Actinobacteria, *Faecalibacterium*, *Turicibacter*, and *Pseudobutyrivibrio*) and alters the colonic metabolite profiles, such as guanidoacetic acid, beta-sitosterol, inosine, cellobiose, indole, and polyamine. Antibiotic supplementation in sow-offspring's diets decreases several beneficial bacteria (including Bacteroidetes, Actinobacteria, *Unclassified_Lachnospiraceae*, and *Prevotella*) and has a favorable effect on improving the fatty acid composition of the LDM to some extent, while presenting the opposite effect on the PMM.

**KEYWORDS** Bama mini-pig, colonic content, lipid metabolism, microbiota composition, probiotics, synbiotics, skeletal muscle

The gut microbiota and its metabolic activities are the key factors for animal health and performance. Accumulating research evidence has demonstrated that gut microbiota could influence feed intake, energy homeostasis, immunity, endocrine systems, and the brain function of the host (1). Moreover, gut microbiota is closely correlated with other factors that influence meat quality, including fiber characteristics and lipid metabolism (2). Previous studies have demonstrated that gut microbiota is the key factor that affects dietary energy gain and storage, and the ratio of Firmicutes to Bacteroidetes is related to energy acquisition and fat deposition of the host (3). In addition, several bacterial genera, such as *Blautia*, *Roseburia*, and *Lactobacillus*, play vital roles in regulating lipid metabolism and fat deposition. Notably, *Prevotella*, *Treponema*, *Bacteroides*, and *Clostridium* positively correlated with intramuscular fat (IMF) and are vital factors influencing meat quality (2). Therefore, manipulating gut microbiota may be a potential strategy to improve meat quality and flavor.

Maternal micro-ecology affects pregnancy, fetal development, and the future health of the offspring piglets (4). The microbiota profile of newborns mainly originates from the maternal amniotic fluid, placenta, gut, and vagina, and microbiota colonization occurs prenatally (5). In addition, the microbiota structure in sows' gut is mainly affected by diet, stress, antibiotic exposure, and the health status of the dam (5). Therefore, modulating sow gut microbiota is an effective strategy to establish beneficial bacteria more rapidly in the gut of newborn piglets. Furthermore, a previous study highlighted that the switch between nursing mother milk and host genetics strongly influenced mammalian gut microbiota development (6). Hence, the suckling period offers a primary window for gut microbiota modulations. As mentioned earlier, regulating maternal and offspring gut microbiota may be a strategy to improve the offspring's growth and meat quality. However, there are few studies available on this aspect at present.

Recently, the conducive functions of gut/intestinal microbiota/bacteria on animal performance and health have gained particular attention (7). As feed additives, the effects of prebiotics, probiotics, and synbiotics have been extensively studied and found to improve the host's growth, production, and health by microbiota-induced mechanisms (8). On the other hand, increased attention has been recently paid to the roles of probiotics and synbiotics in improving meat quality by increasing the redness, tenderness, protein, and IMF content, as well as decreasing drip loss (9, 10). In addition, our previous study indicated that supplementing probiotics or synbiotics in sows and their offspring's diets has beneficial effects on offspring piglet health by enhancing *Bifidobacterium* and *Lactobacillus* abundances and decreasing *Escherichia coli* abundance, as well as enhancing the immunity and antioxidant capacity of offspring piglets (11). However, few studies focused on whether supplementing those probiotics and synbiotics to sow-offspring diets could play positive roles in the gut microbiota composition and metabolism of offspring.

Therefore, we hypothesized that probiotic and synbiotic supplementation to sow-offspring diets could improve lipid metabolism in skeletal muscle by regulating gut microbiota and metabolites of offspring piglets. To test this hypothesis, we determined gene expression related to lipid metabolism in the skeletal muscle and evaluated the colonic microbiome and metabolome of offspring piglets to investigate the effects of

these two additives on sow-offspring's dietary intervention. Furthermore, we examined the potential relationship between lipid metabolism, colonic microbiota, and their metabolites to explore the underlying mechanisms. The findings will contribute to the theoretical foundation for probiotics and synbiotics to improve lipid metabolism through the mother-offspring integration of microbiota modulation.

## MATERIALS AND METHODS

### Animals and diets

The feeding experiment was carried out at the Mini-pig Experiment Station of Goat Chong, Changde, China. Sows were artificially inseminated uniformly, and then, 64 pregnant Bama mini-pigs were selected with similar parity (3–5) and body condition and randomly assigned into four groups: the control, antibiotics, probiotic, and synbiotic groups. The control group was fed an antibiotic-free basal diet. The antibiotic group was fed the basal diet supplemented with 50 mg/kg virginiamycin. The probiotic group was fed the basal diet supplemented with 200 mL/d head probiotics. The synbiotic group was fed the basal diet supplemented with 500 mg/kg xylo-oligosaccharides (XOS) and 200 mL/d head probiotics. The sows received these additives during the gestation and lactation periods. Each group consisted of 16 sows (replicates/pens). Sows were housed in individual pens (2.2 × 0.6 m) during gestation. The sows were fed with 0.8, 1.0, 1.2, 1.5, and 2.0 kg of pregnancy diets from days 1–15, 16–30, 31–75, 76–90, and 91–105 of pregnancy, respectively; fed with 1 kg of lactation diets a week before parturition and *ad libitum* access after 3 days of parturition; and fed with 2.4 kg of a lactation diets until weaning. All sows had free access to water and were fed at 8:00 a.m. and 5:00 p.m. daily. The sows and their piglets were kept in standard farrowing crates with 2.2 × 1.8 m from 7 days before delivery until weaning.

To perform the subsequent feeding trial, a total of 128 piglets [one male and one female, close to the average body weight (BW) of the litter] were selected after weaning (28 days old) and then transferred to the nursery house. Two litters (four piglets in total) from each treatment were combined into one pen, resulting in four piglets per pen and eight pens (replicates) per group after acclimating for one week. These piglets were divided into four treatment groups and had *ad libitum* access to feed at all times. The supplementation levels of probiotics and synbiotics in offspring diets during 35–95 days old were 30 mL/d head probiotic mixture and 250 mg/kg XOS + 30 mL/d head probiotic mixture, respectively, and during 96–125 days old, those were 60 mL/d head probiotic mixture and 250 mg/kg XOS + 60 mL/d head probiotic mixture, respectively. The probiotic mixture was provided by Hunan Lifeng Biotechnology Co. Ltd. (Changsha, China) and contained *Lactobacillus plantarum* B90 (CGMCC1.12934) ≥ 1 × 10$^8$ CFU/mL and *Saccharomyces cerevisiae* P11 (CGMCC2.3854) ≥ 0.2 × 10$^8$ CFU/mL. The XOS (≥35%) was provided by Shandong Longlive Biotechnology Co. Ltd. (Shandong, China) and contained xylobiose (55%), xylotriose (25%), xylotetraose (10%), xylopentose (5%), xylohexaose (3%), and xyloheptaose (2%), which met the feed additive of XOS recommended requirements (GB/T23747-2009). The feeding method of these additives was consistent with our previous study (12). The basal diet composition and nutrient levels for sows and offspring piglets are shown in Tables S1 and S2, respectively. Feeding and management (including vaccination program) for sows and offspring piglets were carried out according to the standard operations of commercial pig farms.

### Sample collection

The offspring piglets were weighed at 65, 95, and 125 days old. After fasting for 12 hours, eight pigs from each group (one pig from each pen with an average BW of the pen) at 65, 95, and 125 days old were selected for sampling. Plasma was obtained from blood samples (10 mL heparinized tubes) collected from the precaval vein. Then, plasma samples were stored at −80℃ to further detect biochemical parameters after

centrifuging at 4°C and 3,500 × *g* for 10 min. After that, pigs were euthanized using electrical stunning (120 V, 200 HZ) and exsanguination. Approximately 2 g of the colon contents (middle section) was collected into sterile centrifuge tubes and then stored at −80°C until further analysis of microbiota and metabolites. Longissimus dorsi muscle (LDM; consists of glycolic muscle fibers) and psoas major muscle (PMM; consists of oxidized muscle fibers) were separated and stored at −20°C and at −80°C for analysis of medium- and long-chain fatty acids (FAs) and mRNA expression, respectively.

## Analysis of medium- and long-chain fatty acids in skeletal muscle

Gas-liquid chromatography (7890A, Agilent, California, USA) was used to detect medium- and long-chain FAs. The samples were performed for pretreatment following the method previously reported by Liu et al. (13). The conditions were consistent with our previous study for GC analysis (14). The retention times of standard reference mixtures (99%, Sigma, St. Louis, MO, USA) were used to identify FAs. The FA contents are expressed as the percentage of the total FAs.

## Analysis of gene expression in skeletal muscle

According to the manufacturer's instructions, the total RNA of the skeletal muscle samples was extracted with available commercial kits (Accurate Biology, Changsha, China). After that, the obtained extracted RNA was reversely transformed into cDNA with commercial gDNA Clean Kits (Accurate Biology, Changsha, China). A LightCycler 480II Real-Time PCR System (Roche, Basel, Switzerland) was performed for the real-time polymerase chain reaction (RT-PCR) analysis to determine mRNA expression. According to the $2^{-\triangle\triangle Ct}$ method, the relative mRNA expression of the target genes was calculated (15). The primers for lipid metabolism related genes were designed and obtained from the Sangon Biotech (Shanghai) Co. Ltd., China (Table S3).

## Analysis of plasma biochemical parameters related to glycolipid metabolism

To measure the plasma levels of amylase (AMS), cholinesterase (CHE), glucose (GLU), high-density lipoprotein-cholesterol (HDL-C), lactic dehydrogenase (LDH), low-density lipoprotein-cholesterol (LDL-C), total cholesterol (TC), and triglyceride (TG), an automatic biochemical analyzer (Cobas c311, F. Hoffmann-La Roche Ltd., Basel, Switzerland) and commercial kits (F. Hoffmann La Roche Ltd., Basel, Switzerland) were used following the instructions of specific kits.

## High-throughput sequencing of colonic microbiota

The Fast DNA SPIN extraction kit (MP Biomedicals, Santa Ana, CA, USA) was used to extract the total bacterial genomic DNA from individual samples of the colonic contents. The quality of DNA samples was checked with gel electrophoresis, and the NanoDrop ND-1000 spectrophotometer instrument was used to measure the DNA concentration of each sample (Thermo Fisher Scientific, Waltham, MA, USA). All bacterial 16S rRNA genes in the hypervariable regions of V3–V4 were amplified according to the method described in a previous study (12). The sequencing of the PCR products was performed on an Illumina Miseq platform (Illumina, San Diego, CA, USA) by the Shanghai Personal Biotechnology Co. Ltd. (Shanghai, China). The 16S sequencing data are deposited in the Science Data Bank with at https://doi.org/10.57760/sciencedb.07263.

Demultiplexing and quality filtering of the raw sequence data were performed on quantitative insights into microbial ecology (QIIME2) (version 2019.4) with slight modifications according to the official tutorials (https://docs.qiime2.org/2019.4/tutorials/). Amplicon sequence variants (ASVs) were obtained using the DADA2 plugin by quality filtering, denoising, merging, and chimera removing (16). Then, non-singleton ASVs were aligned with mafft (17). The ASVs were classified at various taxonomic levels by comparing the sequences with the Greengenes database. The Chao and observed species indices for community richness and the Shannon and Simpson indices for

community diversity were calculated from the normalized ASV reads to estimate the alpha-diversity. The pairwise distances among samples (β-diversity) were assessed using the nonmetric multidimensional scaling (NMDS) analysis. Analysis of similarity (ANOSIM) for multivariate data was performed using the "vegan" package in R (https://www.r-project.org/). The R package was also performed for partial least square discriminant analysis (PLS-DA). Additionally, statistically different bacterial taxa at the phylum and genus levels among different groups were identified using the Kruskal-Wallis test. Linear discriminant analysis effect size (LEfSe) measurements was used to identify the relative richness (LDA > 2, $P < 0.05$) of bacteria and Kyoto Encyclopedia of Genes and Genomes (KEGG) pathways between groups.

## Metabolome profiling in colonic contents

The liquid chromatography-tandem mass spectrometry (LC-MS)/MS technique was applied on a Vanquish ultrahigh-performance LC system (Thermo Fisher, Waltham, MA, USA) coupled with a Q-Orbitrap MS/MS mass spectrometer (Thermo Fisher, Waltham, MA, USA). Briefly, colonic contents were homogenized with two steel balls in the 2 mL EP tube using a tissue crusher. The tissue samples (100 mg) were added to 0.6 mL of methanol (including internal standard), vortexed for 30 s, and centrifuged at 4°C and 12,000 × $g$ for 10 min. Then, the supernatants (200 µL) were used for LC-MS analysis after filtering through a 0.22 µM membrane. The detailed procedures were as described previously (18). The Proteowizard software (v3.0.8789) was used to convert raw data files into mzXML format, and then identification, filtration, and alignment of peaks were processed using the XCMS software (https://www.bioconductor.org/). The mass-to-charge ratio ($m/z$), retention time, and chromatographic data were compared with the internal library to identify differential metabolites. The compound discoverer program was used to annotate the differential metabolites, and the referenced database included the mzCloud database (https://www.mzcloud.org/), as well as BioDeepDB, MetDNA, and MoNA (https://mona.fiehnlab.ucdavis.edu/).

The principal component analysis (PCA) with an unsupervised method and orthogonal partial least squares discriminant analysis (OPLS-DA) with a supervised method were used to obtain the overall structure of the metabolome profiling, general clustering, and trends. The OPLS-DA model was used to calculate the variable importance in projection (VIP). Differences in various differential metabolites among four groups were analyzed by univariate analysis of variance (ANOVA) ($P \leq 0.05$) combined with a statistically significant VIP threshold value (VIP ≥ 1). The MetaboAnalyst 5.0 and KEGG databases (www.kegg.jp) were used to perform pathway analysis of significantly differential metabolites. Heatmaps were constructed using the Euclidian distances and complete linkage grouping with the pheatmap package in R language (https://www.kegg.jp/). The correlations between the significantly differential metabolites were analyzed using Spearman's rank correlation test in the R package.

## Analysis of short-chain fatty acids, indole, skatole, and bioamines of colonic contents

The concentrations of short-chain fatty acids (SCFAs) in the colonic contents were determined by gas chromatography (Agilent 7890A, AgilentInc., Palo Alto, CA, USA), as described in a previous study (19). Based on the previously described method (20), the concentrations of colonic indole, skatole, and bioamine were detected using high-performance liquid chromatography (Agilent 1290, Agilent Inc., Palo Alto, CA, USA).

## Statistical analysis

The medium- and long-chain FA content in skeletal muscles, plasma biochemical parameters, mRNA expression of genes, and colonic metabolite data were analyzed using one-way ANOVA with SPSS 25.0 software (IBM Inc., Chicago, IL, USA) and Tukey's *post-hoc* test. The results are presented as means with their standard error of the mean (SEM). The mRNA expression of colonic gene and metabolite data were visualized using

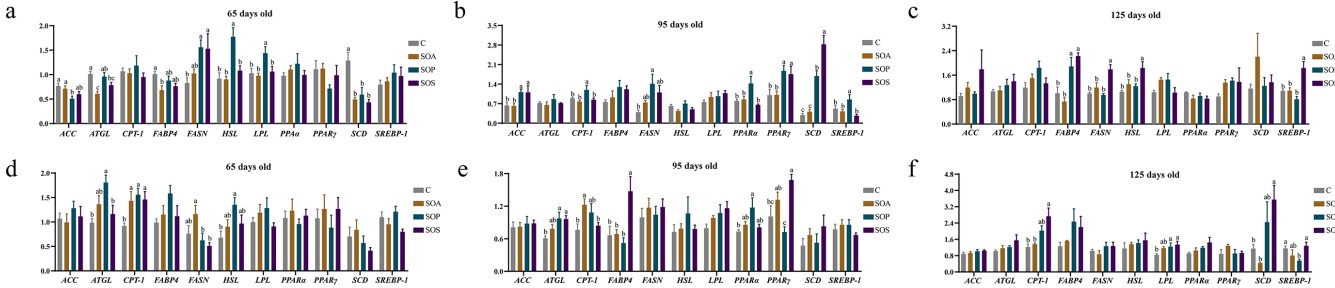

**FIG 1** The mRNA expression of genes related to lipid metabolism in the longissimus dorsi muscle (a–c) and psoas major muscle (d–f) of offspring piglets at 65, 95, and 125 days old. Different lowercase letters indicate a significant difference ($P < 0.05$). C, control group; SOA, antibiotic supplementation in sow-offspring diets; SOP, probiotic supplementation in sow-offspring diets; SOS, synbiotic supplementation in sow-offspring diets; *ACC*, acetyl-CoA carboxylase; *ATGL*, adipose triglyceride lipase; *CPT*-1, carnitine palmityl transferase-1; *FABP4*, fatty acid binding protein 4; *FASN*, fatty acid synthase; *HSL*, hormone sensitive lipase; *LPL*, lipoprotein lipase; *PPARα*, peroxisome proliferator-activated receptor α; *PPARγ*, peroxisome proliferator-activated receptor γ; *SCD*, stearyl coenzyme A desaturase; *SREBP-1*, sterol-regulatory element binding protein-1. The replicates per group at 65 days old were 8. The replicates of the C, SOA, SOP, and SOS groups at 95 days old were 8, 8, 8, and 7, respectively. The replicates of the C, SOA, SOP, and SOS groups at 125 days old were 8, 5, 6, and 6, respectively.

the GraphPad Prism version 8.0 (GraphPad Software, San Diego, CA, USA). Statistical significance was considered based on a $P < 0.05$, and $0.05 \leq P < 0.10$ was considered a trend. Correlations between differential metabolite concentrations and the relative abundance of bacterial taxa at the genus level were determined using Spearman's rank correlation test in the R package.

## RESULTS

### Changes in medium- and long-chain fatty acids in the skeletal muscle of offspring piglets

The changes in FA content in the skeletal muscle of offspring piglets by sow-offspring probiotic and synbiotic supplementation are shown in Tables 1 and 2. In the LDM of offspring piglets, the C12:0 content of the SOA (antibiotic supplementation in sow-off-spring diets), SOP (probiotic supplementation in sow-offspring diets), and SOS (synbiotic supplementation in sow-offspring diets) groups, C22:6n3, PUFA, n-3 PUFA, and n-6 PUFA contents of the SOA and SOP groups, C14:0 content of the SOA and SOS groups, C18:3n3, C18:3n6, C20:3n6, and UFA contents and n-3/n-6 PUFA of the SOA group, and C23:0 content of the SOP group were higher ($P < 0.05$), whereas the C20:0 content of the SOP group was lower ($P < 0.05$) at 65 days old in comparison with that of the C group. The C10:0 content of the SOP and SOS groups, C18:2n6c, C18:3n3, C20:3n6, C20:4n6, PUFA, n-3 PUFA, and PUFA/SFA of the SOA group, and C18:1n9c of the SOP group were higher ($P < 0.05$), whereas C12:0 content of the SOA, SOP, and SOS groups, C24:0 of the SOP and SOS groups, C16:0, C18:1n9c, C20:0, SFA, and n-3/n-6 PUFA of the SOA group, and C18:2n6c content of the SOS group were lower ($P < 0.05$) at 95 days old in comparison with the C group. The C18:0 content of the SOA and SOS groups, C20:0 of the SOA group, and C16:1 of the SOP group were higher ($P < 0.05$), whereas the C18:2n6c, C20:1, and C20:2 contents of the SOA and SOP groups and C15:0 of the SOP and SOS groups were lower ($P < 0.05$) at 125 days old in comparison with the C group (Table 1).

As shown in Table 2, in the PMM of offspring piglets, C18:2n6c, C18:3n6, C24:0, C22:6n3, and n-3 PUFA contents of the SOA group were increased, whereas the IMF content of the SOP group was decreased, as well as C18:1n9t of the SOP and SOS groups, C16:0, C18:0, C20:0, and SFA of the SOP group, and C16:1, C20:3n6, UFA, and PUFA/SFA of the SOS group at 65 days old in comparison with the C group ($P < 0.05$). The IMF content of the SOP group and C18:1n9t content of the SOA and SOP groups were higher, whereas the C20:1, C18:3n3, n-3 PUFA, and n-3/n-6 PUFA ratio of the SOA, SOP, and SOS groups were lower, as well as C16:0, C18:1n9c, and MUFA contents of the SOA group, and C20:3n6 of the SOS group at 95 days old in comparison with the C group ($P <$

**TABLE 1** Effects of probiotic and synbiotic supplementation in sow-offspring diets on intramuscular fat and medium- and long-chain fatty acid content in the longissimus dorsi muscle of offspring piglets (%)[a]

| Item[b] | C group | SOA group | SOP group | SOS group | SEM | P values |
|---|---|---|---|---|---|---|
| Intramuscular fat | | | | | | |
| 65 days old | 2.721 | 2.726 | 2.540 | 2.796 | 0.227 | 0.872 |
| 95 days old | 2.624 | 1.875 | 2.515 | 2.717 | 0.252 | 0.098 |
| 125 days old | 2.148 | 2.975 | 3.054 | 2.820 | 0.422 | 0.363 |
| C10:0 | | | | | | |
| 65 days old | 0.024 | 0.028 | 0.024 | 0.027 | 0.002 | 0.192 |
| 95 days old | 0.022[b] | 0.022[b] | 0.027[a] | 0.027[a] | 0.002 | 0.036 |
| 125 days old | 0.024 | 0.023 | 0.025 | 0.025 | 0.002 | 0.970 |
| C12:0 | | | | | | |
| 65 days old | 0.036[b] | 0.042[a] | 0.045[a] | 0.044[a] | 0.002 | 0.002 |
| 95 days old | 0.041[a] | 0.026[a] | 0.028[a] | 0.034[b] | 0.002 | <0.001 |
| 125 days old | 0.028 | 0.028 | 0.027 | 0.030 | 0.002 | 0.612 |
| C14:0 | | | | | | |
| 65 days old | 0.447[b] | 0.506[a] | 0.472[ab] | 0.511[a] | 0.014 | 0.012 |
| 95 days old | 0.487 | 0.442 | 0.492 | 0.458 | 0.036 | 0.717 |
| 125 days old | 0.497 | 0.448 | 0.453 | 0.517 | 0.026 | 0.231 |
| C14:1 | | | | | | |
| 65 days old | 0.010[b] | 0.012[ab] | 0.013[a] | 0.012[ab] | 0.001 | 0.039 |
| 95 days old | — | — | — | — | — | — |
| 125 days old | — | — | — | — | — | — |
| C15:0 | | | | | | |
| 65 days old | 0.024 | 0.027 | 0.026 | 0.025 | 0.001 | 0.379 |
| 95 days old | 0.022 | 0.023 | 0.021 | 0.021 | 0.002 | 0.713 |
| 125 days old | 0.022[a] | 0.021[a] | 0.015[b] | 0.016[b] | 0.001 | 0.008 |
| C16:0 | | | | | | |
| 65 days old | 8.137 | 8.397 | 8.027 | 8.029 | 0.173 | 0.399 |
| 95 days old | 8.334[a] | 7.675[b] | 8.416[a] | 8.486[a] | 0.176 | 0.010 |
| 125 days old | 8.231 | 8.132 | 8.055 | 8.296 | 0.224 | 0.877 |
| C16:1 | | | | | | |
| 65 days old | 1.177 | 1.266 | 1.240 | 1.133 | 0.045 | 0.162 |
| 95 days old | 0.728 | 0.749 | 0.764 | 0.815 | 0.037 | 0.418 |
| 125 days old | 0.785[b] | 0.775[b] | 0.922[a] | 0.830[ab] | 0.032 | 0.017 |
| C17:0 | | | | | | |
| 65 days old | 0.103 | 0.106 | 0.108 | 0.107 | 0.004 | 0.857 |
| 95 days old | 0.091 | 0.091 | 0.094 | 0.103 | 0.005 | 0.289 |
| 125 days old | 0.082 | 0.085 | 0.063 | 0.077 | 0.007 | 0.148 |
| C18:0 | | | | | | |
| 65 days old | 4.364 | 4.071 | 4.131 | 4.247 | 0.098 | 0.182 |
| 95 days old | 5.141 | 4.885 | 5.080 | 5.068 | 0.119 | 0.463 |
| 125 days old | 4.670[b] | 5.170[a] | 4.438[b] | 4.953[a] | 0.091 | <0.001 |
| C18:1n9c | | | | | | |
| 65 days old | 8.753 | 8.561 | 8.171 | 8.818 | 0.219 | 0.178 |
| 95 days old | 8.820[b] | 8.052[a] | 9.554[a] | 9.235[ab] | 0.208 | <0.001 |
| 125 days old | 9.722 | 9.374 | 9.807 | 9.880 | 0.332 | 0.758 |
| C18:1n9t | | | | | | |
| 65 days old | 0.036 | 0.039 | 0.039 | 0.042 | 0.002 | 0.202 |
| 95 days old | 0.034 | 0.033 | 0.037 | 0.034 | 0.002 | 0.612 |
| 125 days old | 0.042 | 0.040 | 0.040 | 0.035 | 0.002 | 0.117 |
| C18:2n6c | | | | | | |
| 65 days old | 4.135[ab] | 4.578[a] | 4.397[a] | 3.881[b] | 0.140 | 0.009 |
| 95 days old | 3.179[b] | 3.531[a] | 2.988[b] | 2.651[a] | 0.105 | <0.001 |

**TABLE 1** Effects of probiotic and synbiotic supplementation in sow-offspring diets on intramuscular fat and medium- and long-chain fatty acid content in the longissimus dorsi muscle of offspring piglets (%)[a] (Continued)

| Item[b] | C group | SOA group | SOP group | SOS group | SEM | P values |
|---|---|---|---|---|---|---|
| 125 days old | 2.603[a] | 2.017[b] | 1.980[b] | 2.236[ab] | 0.135 | 0.008 |
| C18:3n3 | | | | | | |
| 65 days old | 0.165[b] | 0.188[a] | 0.170[b] | 0.161[b] | 0.004 | <0.001 |
| 95 days old | 0.126[b] | 0.139[a] | 0.127[b] | 0.125[b] | 0.003 | 0.012 |
| 125 days old | — | — | — | — | — | — |
| C18:3n6 | | | | | | |
| 65 days old | 0.023[b] | 0.030[a] | 0.026[ab] | 0.026[ab] | 0.002 | 0.022 |
| 95 days old | — | — | — | — | — | — |
| 125 days old | — | — | — | — | — | — |
| C20:0 | | | | | | |
| 65 days old | 0.063[a] | 0.064[a] | 0.054[b] | 0.066[a] | 0.002 | 0.009 |
| 95 days old | 0.070 | 0.071 | 0.072 | 0.074 | 0.003 | 0.848 |
| 125 days old | 0.063[b] | 0.079[a] | 0.065[b] | 0.068[b] | 0.003 | 0.012 |
| C20:1 | | | | | | |
| 65 days old | 0.204[ab] | 0.237[a] | 0.194[b] | 0.227[ab] | 0.010 | 0.013 |
| 95 days old | 0.303[a] | 0.241[b] | 0.268[ab] | 0.300[a] | 0.011 | 0.002 |
| 125 days old | 0.097[a] | 0.080[b] | 0.081[b] | 0.090[ab] | 0.004 | 0.013 |
| C20:2 | | | | | | |
| 65 days old | 0.163[ab] | 0.170[a] | 0.144[b] | 0.152[ab] | 0.006 | 0.015 |
| 95 days old | 0.153 | 0.147 | 0.139 | 0.142 | 0.007 | 0.469 |
| 125 days old | 0.105[a] | 0.078[a] | 0.074[a] | 0.092[ab] | 0.005 | <0.001 |
| C20:3n6 | | | | | | |
| 65 days old | 0.101 | 0.118 | 0.115 | 0.100 | 0.007 | 0.189 |
| 95 days old | 0.080[b] | 0.098[a] | 0.072[b] | 0.068[b] | 0.006 | 0.005 |
| 125 days old | 0.072 | 0.065 | 0.065 | 0.062 | 0.007 | 0.714 |
| C20:4n6 | | | | | | |
| 65 days old | 0.858 | 1.005 | 0.988 | 0.787 | 0.078 | 0.173 |
| 95 days old | 0.512[b] | 0.803[a] | 0.466[b] | 0.397[b] | 0.057 | <0.001 |
| 125 days old | 0.521 | 0.520 | 0.453 | 0.452 | 0.064 | 0.770 |
| C22:6n3 | | | | | | |
| 65 days old | 0.028[b] | 0.050[a] | 0.045[a] | 0.031[b] | 0.004 | 0.002 |
| 95 days old | — | — | — | — | — | — |
| 125 days old | — | — | — | — | — | — |
| C23:0 | | | | | | |
| 65 days old | 0.021[b] | 0.023[a] | 0.031[a] | 0.024[ab] | 0.002 | 0.049 |
| 95 days old | — | — | — | — | — | — |
| 125 days old | — | — | — | — | — | — |
| C24:0 | | | | | | |
| 65 days old | 0.052 | 0.063 | 0.061 | 0.050 | 0.004 | 0.099 |
| 95 days old | 0.051[a] | 0.059[a] | 0.035[b] | 0.035[b] | 0.003 | <0.001 |
| 125 days old | 0.048 | 0.047 | 0.047 | 0.052 | 0.006 | 0.919 |
| SFA | | | | | | |
| 65 days old | 13.267 | 13.755 | 12.974 | 13.112 | 0.219 | 0.088 |
| 95 days old | 14.218[a] | 13.287[b] | 14.371[a] | 14.466[a] | 0.230 | 0.004 |
| 125 days old | 13.712 | 14.252 | 13.176 | 14.015 | 0.367 | 0.247 |
| UFA | | | | | | |
| 65 days old | 15.777[b] | 16.551[a] | 15.409[b] | 15.450[b] | 0.191 | 0.001 |
| 95 days old | 14.318 | 14.223 | 14.291 | 14.208 | 0.267 | 0.990 |
| 125 days old | 14.061 | 13.863 | 13.871 | 13.756 | 0.255 | 0.833 |
| MUFA | | | | | | |
| 65 days old | 10.144 | 10.107 | 9.595 | 10.330 | 0.257 | 0.236 |

**TABLE 1** Effects of probiotic and synbiotic supplementation in sow-offspring diets on intramuscular fat and medium- and long-chain fatty acid content in the longissimus dorsi muscle of offspring piglets (%)[a] *(Continued)*

| Item[b] | C group | SOA group | SOP group | SOS group | SEM | P values |
|---|---|---|---|---|---|---|
| 95 days old | 9.857[ab] | 9.236[b] | 10.466[a] | 10.371[a] | 0.224 | 0.002 |
| 125 days old | 10.761 | 10.675 | 11.190 | 10.717 | 0.463 | 0.856 |
| PUFA | | | | | | |
| 65 days old | 5.404[b] | 6.472[a] | 5.949[a] | 5.120[b] | 0.186 | <0.001 |
| 95 days old | 3.891[b] | 4.755[a] | 3.835[b] | 3.894[b] | 0.222 | 0.017 |
| 125 days old | 3.300 | 2.680 | 2.572 | 3.039 | 0.250 | 0.151 |
| n-3 PUFA | | | | | | |
| 65 days old | 0.225[a] | 0.283[a] | 0.242[b] | 0.220[a] | 0.005 | <0.001 |
| 95 days old | 0.122[b] | 0.139[a] | 0.127[b] | 0.125[b] | 0.003 | 0.004 |
| 125 days old | — | — | — | — | — | — |
| n-6 PUFA | | | | | | |
| 65 days old | 4.989[b] | 5.998[a] | 5.564[a] | 4.729[b] | 0.184 | <0.001 |
| 95 days old | 3.623[ab] | 4.228[a] | 3.574[ab] | 3.449[b] | 0.195 | 0.039 |
| 125 days old | 3.195 | 2.602 | 2.498 | 2.941 | 0.248 | 0.175 |
| PUFA/SFA | | | | | | |
| 65 days old | 0.408 | 0.450 | 0.460 | 0.389 | 0.021 | 0.079 |
| 95 days old | 0.276[b] | 0.334[a] | 0.271[b] | 0.257[b] | 0.018 | 0.024 |
| 125 days old | 0.246 | 0.227 | 0.22 | 0.217 | 0.029 | 0.872 |
| n-3/n-6 PUFA | | | | | | |
| 65 days old | 0.044[ab] | 0.048[a] | 0.043[a] | 0.047[ab] | 0.001 | 0.003 |
| 95 days old | 0.034[a] | 0.029[b] | 0.036[a] | 0.038[a] | 0.001 | 0.002 |
| 125 days old | — | — | — | — | — | — |

[a]Data are expressed as means with the SEM. Different superscript lowercase letters within the same row not followed by the same letter differ significantly ($P < 0.05$). C, control group; SOA, antibiotic supplementation in sow-offspring diets; SOP, probiotic supplementation in sow-offspring diets; SOS, synbiotic supplementation in sow-offspring diets. The replicates per group at 65 days old were 8. The replicates of the C, SOA, SOP, and SOS groups at 95 days old were 8, 8, 8, and 7, respectively. The replicates of the C, SOA, SOP, and SOS groups at 125 days old were 8, 5, 6, and 6, respectively.

[b]SFA, saturated fatty acid; UFA, unsaturated fatty acid; MUFA, monounsaturated fatty acid; PUFA, polyunsaturated fatty acid. The SFA include C10:0, C12:0, C14:0, C15:0, C16:0, C17:0, C18:0, C20:0, C23:0, and C24:0; the UFA include C14:1, C16:1, C18:1n9c, C18:1n9t, C18:2n6c, C18:3n3, C18:3n6, C20:1, C20:2, C20:3n6, C20:4n6, and C22:6n3; the MUFA include C14:1, C16:1, C18:1n9c, C18:1n9t, and C20:1; the PUFA include C18:2n6c, C18:3n3, C18:3n6, C20:2, C20:3n6, C20:4n6, and C22:6n3; the n-3 PUFA include C18:3n3, C18:3n6, and C22:6n3; the n-6 PUFA include C18:2n6c, C20:3n6, and C20:4n6. "—" indicates that the content of the corresponding index is below the detection limit.

0.05). Moreover, the C20:1 content of the SOA group and C20:4n6 and C24:0 of the SOS group were enhanced, whereas the IMF content of the SOS group, C16:1 content of the SOA group, and C18:0 content of the SOP group were decreased at 125 days old in comparison with the C group ($P < 0.05$).

## Changes in mRNA expression of genes related to lipid metabolism in the skeletal muscle of offspring piglets

The mRNA gene expressions related to lipid metabolism are shown in Fig. 1. In the LDM of offspring piglets, compared with that of the C group, the expressions of fatty acid synthase (*FASN*) of the SOP and SOS groups, as well as hormone-sensitive lipase (*HSL*) and lipoprotein lipase (*LPL*) of the SOS group, were upregulated, whereas stearyl coenzyme A desaturase (*SCD*) of the SOA, SOP, and SOS groups, adipose triglyceride lipase (*ATGL*) of the SOA and SOP groups, fatty acid binding protein 4 (*FABP4*) of the SOA group, and acetyl-CoA carboxylase (*ACC*) of the SOP group were downregulated at 65 days old ($P < 0.05$) (Fig. 1a). At 95 days old, the expressions of peroxisome proliferator-activated receptor γ (*PPARγ*) and *ACC* of the SOP and SOS groups and sterol-regulatory element binding protein-1 (*SREBP-1*), *SCD*, peroxisome proliferator-activated receptor α (*PPARα*), *FASN*, and carnitine palmityl transferase-1 (*CPT-1*) of the SOP group were upregulated, when compared with that of the C group ($P < 0.05$) (Fig. 1b). Furthermore, at 125 days old, the *SREBP-1*, *FASN*, and *HSL* expressions of the SOS group were upregulated ($P < 0.05$) compared with that of the C group (Fig. 1c).

In the PMM of offspring piglets, compared with the C group, the expressions of *CPT-1* of the SOA, SOP, and SOS groups and *ATGL* and *HSL* of the SOP group were upregulated

**TABLE 2** Effects of probiotic or synbiotic supplementation in sow-offspring diets on intramuscular fat and medium- and long-chain fatty acid content in the psoas major muscle of offspring piglets (%)[a]

| Item[b] | C group | SOA group | SOP group | SOS group | SEM | P values |
|---|---|---|---|---|---|---|
| Intramuscular fat | | | | | | |
| 65 days old | 1.498[a] | 1.532[a] | 0.986[b] | 1.864[a] | 0.132 | 0.001 |
| 95 days old | 1.080[b] | 1.102[b] | 1.447[a] | 1.083[b] | 0.080 | 0.006 |
| 125 days old | 1.681[a] | 1.798[a] | 1.812[a] | 1.297[b] | 0.102 | 0.007 |
| C12:0 | | | | | | |
| 65 days old | 0.043[ab] | 0.045[a] | 0.038[b] | 0.041[ab] | 0.002 | 0.043 |
| 95 days old | — | — | — | — | — | — |
| 125 days old | — | — | — | — | — | — |
| C14:0 | | | | | | |
| 65 days old | 0.346 | 0.370 | 0.341 | 0.382 | 0.014 | 0.171 |
| 95 days old | 0.311 | 0.279 | 0.310 | 0.323 | 0.014 | 0.185 |
| 125 days old | 0.437 | 0.399 | 0.419 | 0.408 | 0.020 | 0.557 |
| C15:0 | | | | | | |
| 65 days old | 0.036 | 0.035 | 0.035 | 0.035 | 0.003 | 0.994 |
| 95 days old | 0.030 | 0.029 | 0.028 | 0.030 | 0.001 | 0.641 |
| 125 days old | — | — | — | — | — | — |
| C16:0 | | | | | | |
| 65 days old | 7.220[a] | 7.384[a] | 6.732[b] | 7.171[a] | 0.085 | <0.001 |
| 95 days old | 6.600[a] | 6.161[b] | 6.502[a] | 6.611[a] | 0.115 | 0.032 |
| 125 days old | — | — | — | — | — | — |
| C16:1 | | | | | | |
| 65 days old | 0.916[a] | 0.922[a] | 0.929[a] | 0.822[b] | 0.026 | 0.022 |
| 95 days old | 0.529 | 0.514 | 0.583 | 0.550 | 0.029 | 0.377 |
| 125 days old | 0.654[ab] | 0.578[b] | 0.722[a] | 0.597[b] | 0.030 | 0.016 |
| C17:0 | | | | | | |
| 65 days old | 0.113 | 0.111 | 0.122 | 0.114 | 0.007 | 0.667 |
| 95 days old | 0.073 | 0.084 | 0.078 | 0.089 | 0.004 | 0.051 |
| 125 days old | 0.108 | 0.137 | 0.104 | 0.107 | 0.008 | 0.057 |
| C18:0 | | | | | | |
| 65 days old | 4.300[ab] | 4.460[a] | 3.995[a] | 4.236[b] | 0.063 | <0.001 |
| 95 days old | 4.507 | 4.470 | 4.515 | 4.451 | 0.076 | 0.927 |
| 125 days old | 5.032[ab] | 5.170[ab] | 4.925[a] | 5.296[a] | 0.065 | 0.003 |
| C18:1n9c | | | | | | |
| 65 days old | 6.786[a] | 6.678[a] | 6.014[b] | 6.859[a] | 0.171 | 0.006 |
| 95 days old | 6.222[a] | 5.545[b] | 6.235[a] | 6.530[a] | 0.119 | <0.001 |
| 125 days old | 8.759[ab] | 9.384[ab] | 9.584[a] | 8.493[b] | 0.250 | 0.018 |
| C18:1n9t | | | | | | |
| 65 days old | 0.035[a] | 0.033[ab] | 0.028[b] | 0.030[b] | 0.001 | 0.008 |
| 95 days old | 0.028[b] | 0.035[a] | 0.034[a] | 0.031[ab] | 0.001 | 0.001 |
| 125 days old | 0.039 | 0.037 | 0.036 | 0.037 | 0.003 | 0.884 |
| C18:2n6c | | | | | | |
| 65 days old | 5.290[ab] | 6.505[a] | 5.719[b] | 4.918[a] | 0.180 | <0.001 |
| 95 days old | 5.221 | 5.192 | 4.735 | 4.739 | 0.176 | 0.091 |
| 125 days old | 4.752 | 4.487 | 4.745 | 5.044 | 0.229 | 0.463 |
| C18:3n3 | | | | | | |
| 65 days old | 0.151[ab] | 0.162[a] | 0.151[ab] | 0.144[b] | 0.003 | 0.005 |
| 95 days old | 0.111 | — | — | — | — | — |
| 125 days old | 0.125 | 0.113 | 0.110 | 0.117 | 0.005 | 0.108 |
| C18:3n6 | | | | | | |
| 65 days old | — | — | — | — | — | — |
| 95 days old | 0.036 | 0.036 | 0.036 | 0.033 | 0.002 | 0.354 |

(*Continued on next page*)

**TABLE 2** Effects of probiotic or synbiotic supplementation in sow-offspring diets on intramuscular fat and medium- and long-chain fatty acid content in the psoas major muscle of offspring piglets (%)[a] (*Continued*)

| Item[b] | C group | SOA group | SOP group | SOS group | SEM | P values |
|---|---|---|---|---|---|---|
| 125 days old | — | — | — | — | — | — |
| C20:0 | | | | | | |
| 65 days old | 0.050[a] | 0.052[a] | 0.038[b] | 0.049[a] | 0.002 | 0.001 |
| 95 days old | 0.046 | 0.045 | 0.047 | 0.050 | 0.001 | 0.096 |
| 125 days old | 0.058 | 0.065 | 0.061 | 0.061 | 0.002 | 0.163 |
| C20:1 | | | | | | |
| 65 days old | 0.165[ab] | 0.184[a] | 0.146[b] | 0.171[ab] | 0.007 | 0.009 |
| 95 days old | 0.187[a] | 0.108[b] | 0.107[b] | 0.110[b] | 0.008 | <0.001 |
| 125 days old | 0.274[b] | 0.337[a] | 0.303[ab] | 0.263[b] | 0.017 | 0.035 |
| C20:2 | | | | | | |
| 65 days old | 0.147[ab] | 0.154[a] | 0.131[b] | 0.138[ab] | 0.005 | 0.020 |
| 95 days old | 0.133 | 0.138 | 0.124 | 0.117 | 0.006 | 0.141 |
| 125 days old | 0.120 | 0.122 | 0.109 | 0.129 | 0.006 | 0.229 |
| C20:3n6 | | | | | | |
| 65 days old | 0.155[b] | 0.176[a] | 0.141[ab] | 0.131[a] | 0.006 | <0.001 |
| 95 days old | 0.151[ab] | 0.176[a] | 0.162[ab] | 0.139[b] | 0.008 | 0.017 |
| 125 days old | 0.139 | 0.143 | 0.143 | 0.151 | 0.011 | 0.880 |
| C20:4n6 | | | | | | |
| 65 days old | 1.596 | 1.779 | 1.534 | 1.298 | 0.117 | 0.054 |
| 95 days old | 1.696 | 1.732 | 1.559 | 1.455 | 0.110 | 0.296 |
| 125 days old | 1.294[b] | 1.257[b] | 1.254[b] | 1.757[a] | 0.077 | <0.001 |
| C22:6n3 | | | | | | |
| 65 days old | 0.056[b] | 0.085[a] | 0.062[b] | 0.050[b] | 0.005 | <0.001 |
| 95 days old | 0.049[ab] | 0.047[ab] | 0.034[b] | 0.062[a] | 0.004 | 0.001 |
| 125 days old | — | — | — | — | — | — |
| C24:0 | | | | | | |
| 65 days old | 0.094[b] | 0.110[a] | 0.087[bb] | 0.081[b] | 0.005 | 0.001 |
| 95 days old | 0.091 | 0.094 | 0.087 | 0.098 | 0.005 | 0.441 |
| 125 days old | 0.109[b] | 0.131[ab] | 0.136[ab] | 0.147[a] | 0.008 | 0.018 |
| SFA | | | | | | |
| 65 days old | 12.241[a] | 12.536[a] | 11.632[b] | 11.987[ab] | 0.166 | 0.005 |
| 95 days old | 11.722[ab] | 11.178[b] | 11.564[ab] | 11.850[a] | 0.162 | 0.039 |
| 125 days old | 5.727 | 5.931 | 5.624 | 5.984 | 0.122 | 0.165 |
| UFA | | | | | | |
| 65 days old | 15.643[ab] | 15.911[a] | 14.997[ab] | 14.739[a] | 0.238 | 0.005 |
| 95 days old | 13.924 | 13.585 | 13.543 | 13.386 | 0.197 | 0.293 |
| 125 days old | 19.027 | 18.696 | 18.882 | 19.328 | 0.321 | 0.602 |
| MUFA | | | | | | |
| 65 days old | 8.282[a] | 7.792[a] | 7.038[b] | 8.001[a] | 0.251 | 0.011 |
| 95 days old | 6.796[a] | 6.205[b] | 6.965[a] | 7.027[a] | 0.165 | 0.006 |
| 125 days old | 12.368 | 12.613 | 12.587 | 11.731 | 0.549 | 0.663 |
| PUFA | | | | | | |
| 65 days old | 7.769[ab] | 8.455[a] | 7.748[ab] | 6.625[b] | 0.329 | 0.005 |
| 95 days old | 7.138 | 7.234 | 6.646 | 6.359 | 0.279 | 0.118 |
| 125 days old | 6.533 | 5.970 | 5.911 | 6.729 | 0.485 | 0.568 |
| n-3 PUFA | | | | | | |
| 65 days old | 0.207[b] | 0.235[a] | 0.211[b] | 0.198[b] | 0.007 | 0.005 |
| 95 days old | 0.132[a] | 0.070[b] | 0.063[b] | 0.055[b] | 0.007 | <0.001 |
| 125 days old | 0.126 | 0.113 | 0.110 | 0.119 | 0.006 | 0.233 |
| n-6 PUFA | | | | | | |
| 65 days old | 7.416[ab] | 8.067[a] | 7.400[ab] | 6.298[b] | 0.321 | 0.005 |

*(Continued on next page)*

**TABLE 2** Effects of probiotic or synbiotic supplementation in sow-offspring diets on intramuscular fat and medium- and long-chain fatty acid content in the psoas major muscle of offspring piglets (%)[a] (*Continued*)

| Item[b] | C group | SOA group | SOP group | SOS group | SEM | P values |
|---|---|---|---|---|---|---|
| 95 days old | 6.882 | 7.261 | 6.478 | 6.218 | 0.281 | 0.072 |
| 125 days old | 6.136 | 5.848 | 5.807 | 6.600T | 0.447 | 0.598 |
| PUFA/SFA | | | | | | |
| 65 days old | 0.623[a] | 0.671[a] | 0.676[a] | 0.539[b] | 0.024 | 0.001 |
| 95 days old | 0.620[ab] | 0.674[a] | 0.575[b] | 0.564[b] | 0.020 | 0.003 |
| 125 days old | 1.087 | 1.008 | 1.102 | 1.121 | 0.086 | 0.835 |
| n-3/n-6 PUFA | | | | | | |
| 65 days old | 0.028 | 0.028 | 0.028 | 0.031 | 0.001 | 0.083 |
| 95 days old | 0.020[a] | 0.010[b] | 0.010[b] | 0.008[b] | 0.001 | <0.001 |
| 125 days old | 0.020 | 0.020 | 0.019 | 0.019 | 0.001 | 0.915 |

[a]Data are expressed as means with the SEM. Different superscript lowercase letters within the same row not followed by the same letter differ significantly ($P < 0.05$). C, control group; SOA, antibiotic supplementation in sow-offspring diets; SOP, probiotic supplementation in sow-offspring diets; SOS, synbiotic supplementation in sow-offspring diets. The replicates per group at 65 days old were 8. The replicates of the C, SOA, SOP, and SOS groups at 95 days old were 8, 8, 8, and 7, respectively. The replicates of the C, SOA, SOP, and SOS groups at 125 days old were 8, 5, 6, and 6, respectively.

[b]SFA, saturated fatty acid; UFA, unsaturated fatty acid; MUFA, monounsaturated fatty acid; PUFA, polyunsaturated fatty acid. The SFA include C12:0, C14:0, C15:0, C16:0, C17:0, C18:0, C20:0, and C24:0; the UFA include C16:1, C18:1n9c, C18:1n9t, C18:2n6c, C18:3n3, C18:3n6, C20:1, C20:2, C20:3n6, C20:4n6, and C22:6n3; the MUFA include C16:1, C18:1n9c, C18:1n9t, and C20:1; the PUFA include C18:2n6c, C18:3n3, C20:2, C20:3n6, C20:4n6, and C22:6n3; the n-3 PUFA include C18:3n3, C18:3n6, and C22:6n3; the n-6 PUFA include C18:2n6c, C20:3n6, and C20:4n6. "—" indicates that the content of the corresponding index is below the detection limit.

($P < 0.05$) at 65 days old (Fig. 1d). At 95 days old, the expressions of *ATGL* of the SOP and SOS groups, *CPT-1* of the SOA group, *PPARα* of the SOP group, and *PPARγ* and *FASN* of the SOS group were elevated ($P < 0.05$) compared with that of the C group (Fig. 1e). The expressions of *LPL* of the SOP and SOS groups and *SCD* and *CPT-1* of the SOS group were elevated ($P < 0.05$), whereas *SREBP-1* of the SOP group was reduced ($P < 0.05$) at 125 days old, when compared with the C group (Fig. 1f).

## Changes in plasma biochemical parameters related to glycolipid metabolism in offspring piglets

The results of plasma biochemical parameters are presented in Table 3. The SOA, SOP, and SOS groups had lower AMS level, whereas the SOP and SOS groups had higher HDL-C level, and the SOS group had a higher TC level at 65 days old than the C group ($P < 0.05$). The LDH, TG, and GLU levels were reduced in the SOA, SOP, and SOS groups, as well as the HDL-C level in the SOA and SOS groups, whereas the CHE level was elevated in the SOP and SOS groups at 95 days old when compared with that in the C group ($P < 0.05$). The SOA, SOP, and SOS groups had higher LDL-C level, the SOA group had a higher GLU level, and the SOS group had a higher TC level, whereas the SOA group had a lower LDH level at 125 days old than the C group ($P < 0.05$)

## Changes in colonic microbiota diversity of offspring piglets

To investigate the impacts of dietary probiotic and synbiotic addition on offspring piglets' colonic microbiota composition, the V3–V4 hypervariable region of the bacterial 16S rRNA gene was amplified and sequenced for each sample using an Illumina Miseq technology. A total of 8,751,055 high-quality sequences were obtained from 88 samples (including 32, 31, and 25 samples at 65, 95, and 125 days old, respectively) with an average length of 412 bp and used for subsequent analysis. The results of rarefaction curves showed that the samples in each group provided sufficient operational taxonomic unit (OTU) coverage (Fig. S1). There were no significant changes ($P > 0.05$) in the α-diversity, including richness estimators (Chao1), diversity indices (Shannon and Simpson), and evenness index (Pielou's) in response to dietary probiotic and synbiotic supplementation (Table S4). The colonic microbial communities had no significant difference, which could be visualized by the NMDS ordination score plots (Fig. 2a through c ; ANOSIM, $P > 0.05$). In addition, the distinct separations of colonic microbial communities among the four groups at 65, 95, and 125 days old are visualized by PLS-DA (Fig. 2d through f).

**TABLE 3** Effects of probiotic or synbiotic supplementation in sow-offspring diets on plasma biochemical parameters of offspring piglets[a]

| Items[b] | C group | SOA group | SOP group | SOS group | SEM | P values |
|---|---|---|---|---|---|---|
| AMS, U/L | | | | | | |
| 65 days old | 2260.50[a] | 1563.00[a] | 1906.29[b] | 1869.83[b] | 66.870 | <0.001 |
| 95 days old | 2186.71 | 2129.00 | 2290.40 | 2098.60 | 62.507 | 0.167 |
| 125 days old | 2306.00[ab] | 2632.50[a] | 2297.25[b] | 2018.25[b] | 108.162 | 0.011 |
| CHE, g/L | | | | | | |
| 65 days old | 534.63 | 514.83 | 526.33 | 551.75 | 16.598 | 0.465 |
| 95 days old | 516.25[a] | 451.17[a] | 379.33[a] | 421.67[ab] | 19.366 | <0.001 |
| 125 days old | 467.67 | 483.75 | 453.75 | 443.00 | 22.977 | 0.663 |
| LDH, U/L | | | | | | |
| 65 days old | 416.75 | 433.29 | 436.17 | 491.80 | 24.709 | 0.180 |
| 95 days old | 465.71[a] | 366.57[b] | 387.50[b] | 382.83[b] | 16.859 | 0.001 |
| 125 days old | 475.57[a] | 353.00[b] | 392.20[ab] | 423.80[ab] | 28.441 | 0.034 |
| HDL-C, mmol/L | | | | | | |
| 65 days old | 0.76[b] | 0.76[b] | 0.94[a] | 0.90[a] | 0.039 | 0.003 |
| 95 days old | 1.07[a] | 0.86[b] | 0.98[a] | 0.84[b] | 0.036 | <0.001 |
| 125 days old | 1.10[b] | 0.89[ab] | 1.02[ab] | 1.32[a] | 0.046 | <0.001 |
| LDL-C, mmol/L | | | | | | |
| 65 days old | 1.28 | 1.32 | 1.41 | 1.41 | 0.056 | 0.270 |
| 95 days old | 1.57 | 1.47 | 1.62 | 1.47 | 0.079 | 0.438 |
| 125 days old | 1.53[b] | 1.87[a] | 1.83[a] | 1.98[a] | 0.054 | <0.001 |
| TC, mmol/L | | | | | | |
| 65 days old | 2.37[b] | 2.41[b] | 2.44[b] | 2.64[a] | 0.050 | 0.004 |
| 95 days old | 2.68 | 2.56 | 2.78 | 2.77 | 0.068 | 0.092 |
| 125 days old | 2.78[b] | 2.95[b] | 2.77[b] | 3.27[a] | 0.091 | 0.002 |
| TG, mmol/L | | | | | | |
| 65 days old | 0.49 | 0.57 | 0.55 | 0.49 | 0.029 | 0.122 |
| 95 days old | 0.76[a] | 0.47[b] | 0.47[b] | 0.52[b] | 0.031 | <0.001 |
| 125 days old | 0.62 | 0.57 | 0.74 | 0.59 | 0.047 | 0.097 |
| GLU, mmol/L | | | | | | |
| 65 days old | 5.30 | 5.53 | 5.20 | 5.10 | 0.238 | 0.615 |
| 95 days old | 6.40[a] | 4.75[bb] | 4.92[b] | 4.96[b] | 0.265 | <0.001 |
| 125 days old | 4.48[b] | 5.18[a] | 4.60[b] | 4.78[b] | 0.133 | 0.010 |

[a]Data are expressed as means with the SEM. Different superscript lowercase letters within the same row not followed by the same letter differ significantly ($P < 0.05$). C, control group; SOA, antibiotic supplementation in sow-offspring diets; SOP, probiotic supplementation in sow-offspring diets; SOS, synbiotic supplementation in sow-offspring diets. The replicates per group at 65 days old were 8. The replicates of the C, SOA, SOP, and SOS groups at 95 days old were 8, 8, 8, and 7, respectively. The replicates of the C, SOA, SOP, and SOS groups at 125 days old were 8, 5, 6, and 6, respectively.

[b]AMS, amylase; CHE, cholinesterase; LDH, lactic dehydrogenase; LDL-C, low-density lipoprotein-cholesterol; HDL-C, high-density lipoprotein-cholesterol; TC, total cholesterol; TG, triglyceride; GLU, glucose. C, control group; SOA, antibiotic supplementation in sow-offspring diets; SOP, probiotic supplementation in sow-offspring diets; SOS, synbiotic supplementation in sow-offspring diets.

## Changes in overall colonic microbiota structure of offspring piglets

To further determine colonic microbiota that were responsible for the alteration due to the dietary antibiotic, probiotic, and synbiotic treatments, the microbiota profile was evaluated at the phylum and genus levels. Firmicutes, Bacteroidetes, and Spirochaetes were the most dominant phyla at 65, 95, and 125 days old (Fig. 3a through c). These three phyla in the C, SOA, SOP, and SOS groups accounted for 95.43%, 96.56%, 96.27%, and 95.68% of the total reads at 65 days old; 97.12%, 95.44%, 96.23%, and 96.03% of the total reads at 95 days old; and 95.43%, 96.56%, 96.27%, and 95.68% of the total reads at 125 days old, respectively. The top 10 abundant genera were *Lactobacillus*, *Streptococcus*, *Gemmiger*, *Treponema*, *unclassified_Clostridiales*, *Ruminococcus*, *SMB53*, *Oscillospira*, *Prevotella*, and *Clostridium* at 65 days old (Fig. 3g); *Lactobacillus*, *Treponema*, *unclassified_Clostridiales*, *Oscillospira*, *SMB53*, *Clostridium*, *Ruminococcus*, *Prevotella*, *Gemmiger*, and *unclassified_Lachnospiraceae* at 95 days old (Fig. 3h); and *Lactobacillus*, *unclassified_Clostridiales*, *Treponema*, *SMB53*, *Oscillospira*, *Prevotella*, *Ruminococcus*, *Turicibacter*,

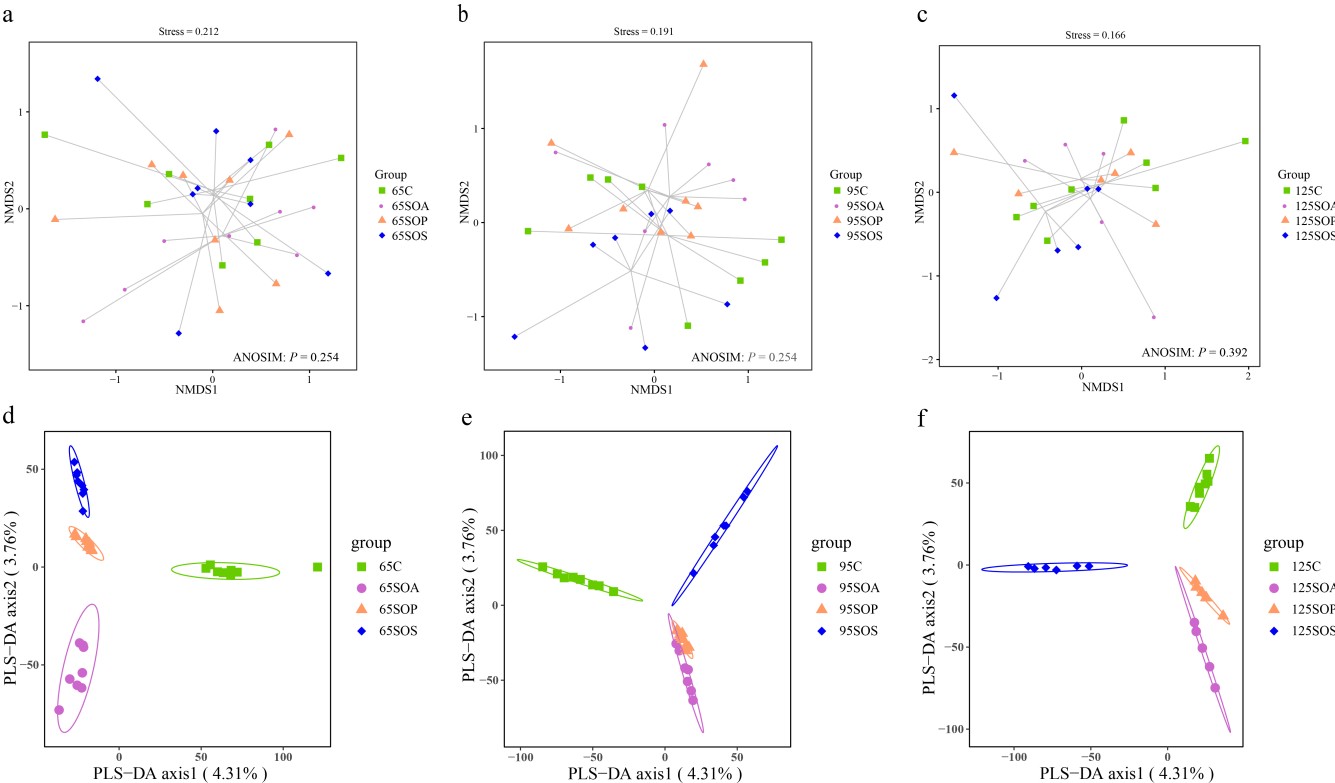

**FIG 2** NMDS ordination plots based on the Bray-Curtis distance metric (a, b, and c) and partial least square discriminant analysis (PLS-DA) plots (d, e, and f) of the colonic microbiota community of offspring piglets at 65, 95, and 125 days old. C, control group; SOA, antibiotic supplementation in sow-offspring diets; SOP, probiotic supplementation in sow-offspring diets; SOS, synbiotic supplementation in sow-offspring diets. The replicates per group at 65 days old were 8. The replicates of the C, SOA, SOP, and SOS groups at 95 days old were 8, 8, 8, and 7, respectively. The replicates of the C, SOA, SOP, and SOS groups at 125 days old were 8, 5, 6, and 6, respectively.

*Clostridium*, and *Gemmiger* at 125 days old (Fig. 3i). *Lactobacillus* was the most dominant genera in the C, SOA, SOP, and SOS groups and at three different ages. Moreover, the common genera at three different ages were *Lactobacillus*, *Gemmiger*, *Treponema*, *unclassified_Clostridiales*, *Ruminococcus*, *SMB53*, *Oscillospira*, *Prevotella*, and *Clostridium*.

The Kruskal-Wallis test was used to determine the significant differences in colonic microbiota at the phylum and genus levels among different treatment groups. At the phylum level, Actinobacteria abundance was enriched ($P < 0.05$) in the SOA and SOP groups at 65 days old when compared with the C group (Fig. 3d). At 95 days old, the Verrucomicrobia abundance in the SOS group displayed an enhancing trend ($P = 0.055$) compared with the C group (Fig. 3e). At 125 days old, Actinobacteria abundance was reduced in the SOA, SOP, and SOS groups, whereas the Verrucomicrobia abundance was enhanced in the SOA and SOS groups in comparison with the C group ($P < 0.05$) (Fig. 3f). Moreover, at 125 days old, the Bacteroidetes ($P = 0.073$) and Proteobacteria ($P = 0.099$) abundances displayed a decreasing trend in the SOA group than the C, SOP, and SOS groups (Fig. 3f). At the genus level, *Faecalibacterium* abundance displayed an enhancing trend ($P = 0.098$) in the SOP group at 65 days old compared with the C, SOA, and SOS groups (Fig. 3j). The SOA group had a lower *unclassified_Lachnospiraceae* abundance than the C group at 95 days old ($P < 0.05$) (Fig. 3k). At 125 days old, the *Prevotella* abundance had a decreasing trend ($P = 0.090$), whereas the *02d06* abundance had an enhancing trend ($P = 0.090$) in the SOA and SOP groups in comparison with the C and SOS groups. Moreover, *Turicibacter* abundance was enhanced ($P < 0.05$) in the SOP group compared with the SOS group (Fig. 3l).

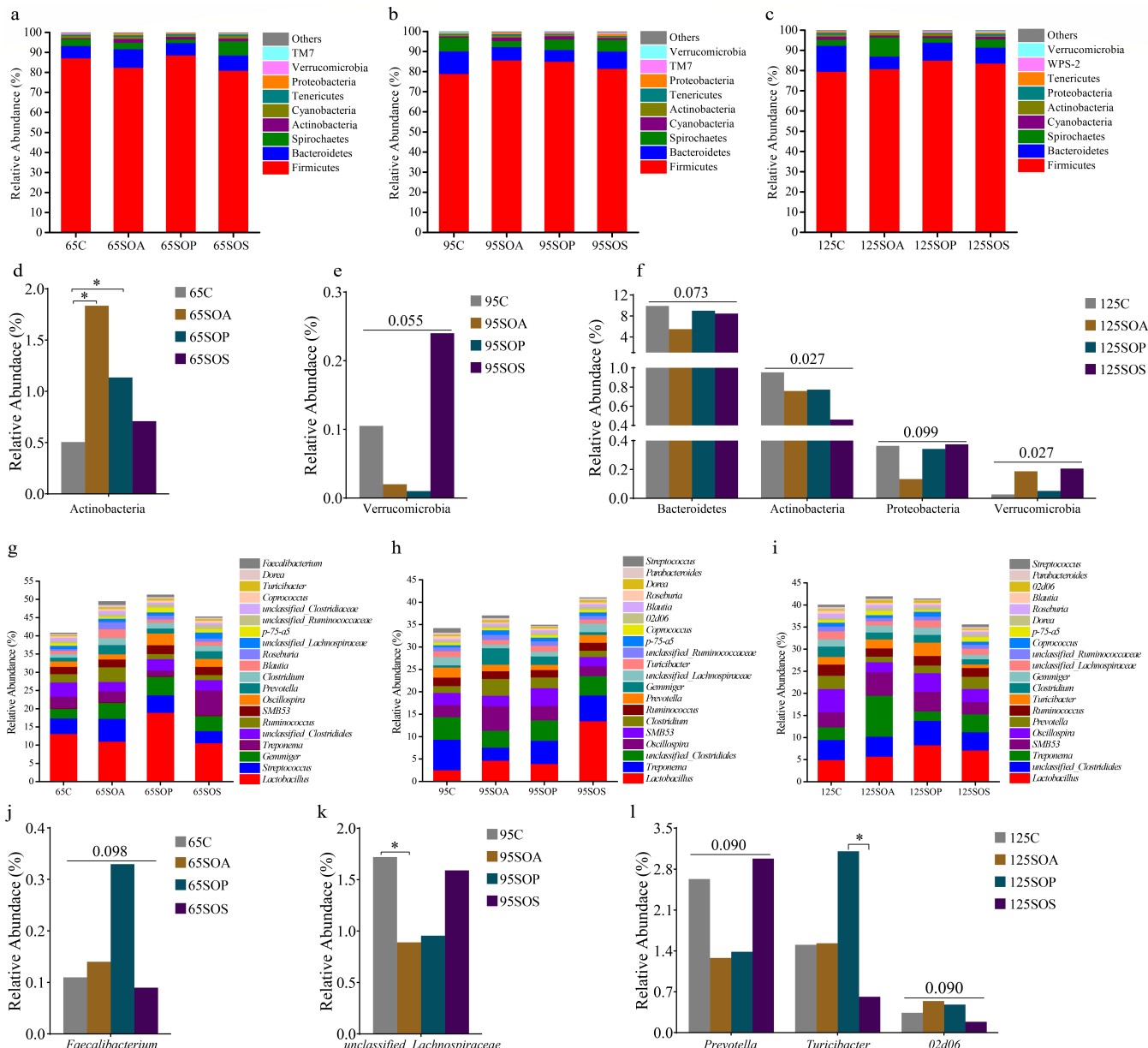

**FIG 3** Relative abundance of colonic microbiota at the phylum (relative abundance > 0.1%) (a–c) and genus (the top 20 genera) (g–i) levels of offspring piglets. The significant changes of phyla at 65 (d), 95 (e), and 125 (f) days old and genera at 65 (j), 95 (k), and 125 (l) days old are presented. The values are expressed as the median. Statistical differences are calculated by the Kruskal-Wallis test: significance is considered at $P$ values < 0.05, and the tendency is considered at $P$ values between 0.05 and 0.1, * $P$ < 0.05. C, control group; SOA, antibiotic supplementation in sow-offspring diets; SOP, probiotic supplementation in sow-offspring diets; SOS, synbiotic supplementation in sow-offspring diets. The replicates per group at 65 days old were 8. The replicates of the C, SOA, SOP, and SOS groups at 95 days old were 8, 8, 8, and 7, respectively. The replicates of the C, SOA, SOP, and SOS groups at 125 days old were 8, 5, 6, and 6, respectively.

## Changes in colonic microbiota communities of offspring piglets

The clustering heatmap was constructed with the abundances of the top 50 bacterial genera to analyze the situation of species abundance distribution in four treatment groups. The heatmap showed that there were differences in enriched genera among the four treatment groups at three different ages (Fig. S2).

LEfSe analysis of the taxa combined with LDA scores ≥ 2 was used to identify the characteristics of bacteria species in different treatments. Firmicutes was significantly enriched ($P$ < 0.05) in the C group at 65 days old, as well as in the SOS group at 95 days

old at the phylum level (Fig. 4a and b). The SOA group had a higher ($P < 0.05$) Actinobacteria abundance at 65 and 95 days old (Fig. 4a and b), and the SOS group had higher ($P < 0.05$) Elusimicrobia abundance at 65 days old and Verrucomicrobia abundance at 125 days old (Fig. 4a and c ), and the SOP group had a higher ($P < 0.05$) Fibrobacter abundance at 125 days old (Fig. 4c). At the genus level, the *02d06* abundance was significantly enriched ($P < 0.05$) in the C group, as well as *Collinsella* and *Sphingomonas* abundances in the SOA group, and *Pseudobutyrivibrio* and *RFN20* abundances in the SOS group at 65 days old (Fig. 4a). The SOA group had higher ($P < 0.05$) *Butyricicoccus* and *Weissella* abundances than the C, SOP, and SOS groups at 95 days old (Fig. 4b). The *02d06* abundance in the SOA group and *Turicibacter*, *Peptococcus*, and *Fibrobacter* abundances in the SOP group were enriched ($P < 0.05$) at 125 days old when compared with the C group (Fig. 4c).

## Predicted functions of colonic microbiota of offspring piglets

The composition of the microbiota community of the KEGG pathway was predicted by a PICRUSt2 approach. At level 1, at 65, 95, and 125 days old, the pathways affiliated with metabolism were 76.19%, 75.61%, and 75.75%, respectively; the pathways affiliated with genetic information processing were 14.75%, 14.66%, and 14.82%, respectively; the pathways involved with cellular processes were 5.44%, 6.21%, and 6.02%, respectively; and the pathways that belonged to environmental information processing were 2.90%, 2.80%, and 2.69% (Fig. S3), respectively.

There were 31, 30, and 31 KEGG pathways enriched at 65, 95, and 125 days old in the colon at level 2 (Fig. S3). These pathways were related to carbohydrate metabolism, amino acid metabolism, metabolism of cofactors and vitamins, metabolism of terpenoids and polyketides, metabolism of other amino acids, replication and repair, lipid metabolism, and energy metabolism at 65, 95, and 125 days old (Fig. S3). After that, the functional pathways among different treatments were evaluated (Fig. 5). The results indicated that the relative abundance of glycan biosynthesis and metabolism was higher ($P < 0.05$) in the SOA group than in the C group at 65 days old (Fig. 5a).

Furthermore, the differences in identified pathways were further clarified at level 3 and identified two, four, and four enriched pathways at 65, 95, and 125 days old (Fig. 5). At 65 days old, the renin-angiotensin system pathway was enriched ($P < 0.05$) in the C group, as well as the N-glycan biosynthesis in the SOA group (Fig. 5a). At 95 days old, steroid biosynthesis and other glycan degradation were enriched in the C group ($P < 0.05$), as well as the toxoplasmosis in the SOA group, and sphingolipid metabolism in the SOS group (Fig. 5b). At 125 days old, D-arginine and D-ornithine metabolism and lipopolysaccharide biosynthesis were enriched ($P < 0.05$) in the C group, as well as

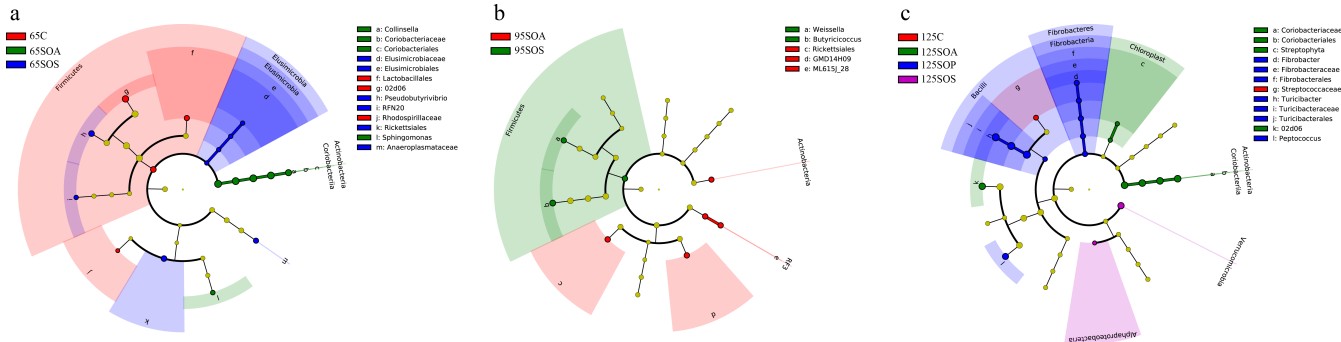

**FIG 4** Differential enrichment of colonic microbiota. Analysis of taxonomic abundance using linear discriminant analysis effect size (LEfSe) (LDA score ≥ 2, $P <$ 0.05) in the colonic contents of offspring piglets at 65 (a), 95 (b), and 125 (c) days old. The cladogram shows the microbial species with a significant difference between the experiment group and the control group in LEfSe analysis. C, control group; SOA, antibiotic supplementation in sow-offspring diets; SOP, probiotic supplementation in sow-offspring diets; SOS, synbiotic supplementation in sow-offspring diets. The replicates per group at 65 days old were 8. The replicates of the C, SOA, SOP, and SOS groups at 95 days old were 8, 8, 8, and 7, respectively. The replicates of the C, SOA, SOP, and SOS groups at 125 days old were 8, 5, 6, and 6, respectively.

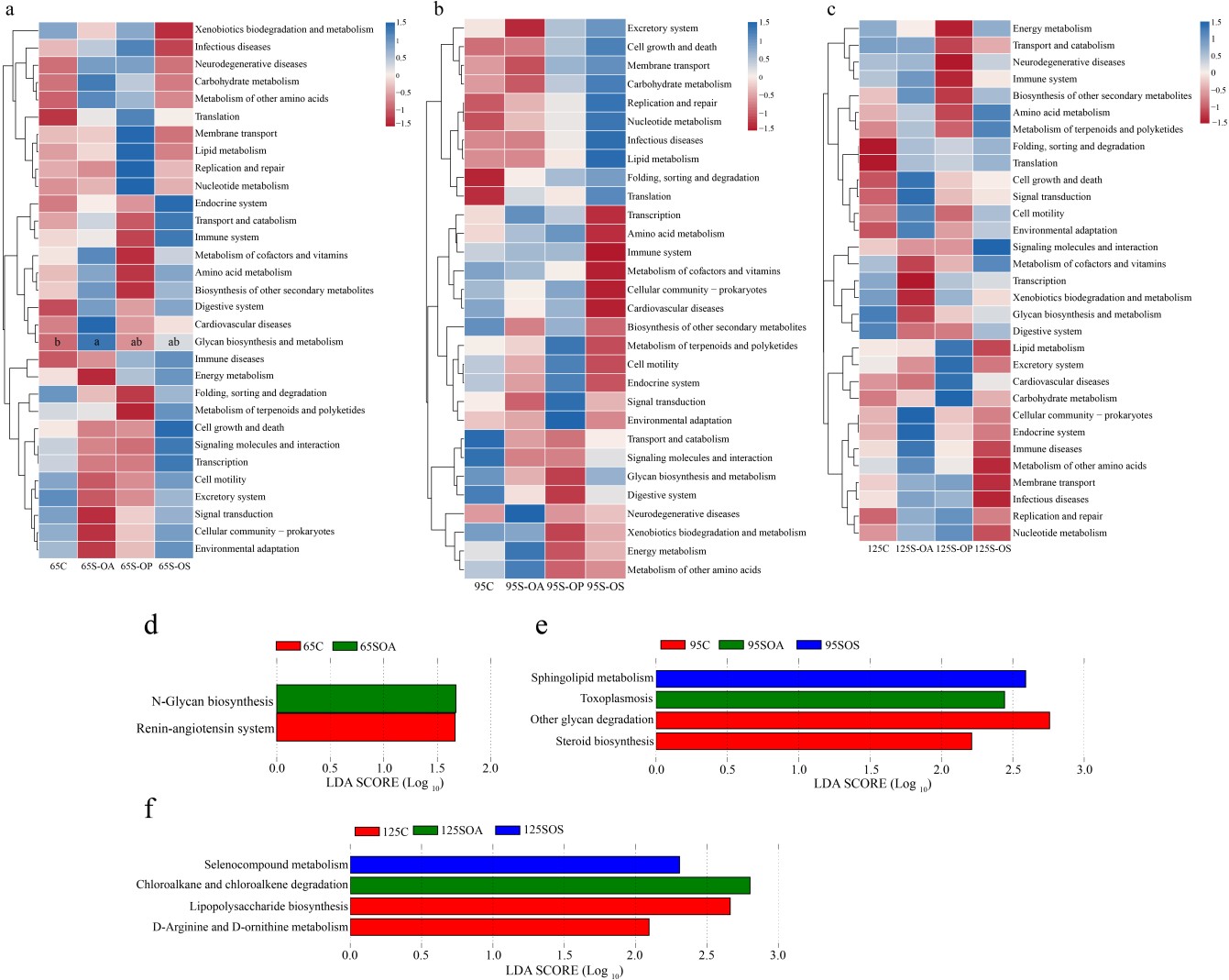

**FIG 5** Comparisons of the microbiota pathways among different treatment groups at 65 (a), 95 (b), and 125 (c) days old at level 2 using Kruskal-Wallis test. Predicted function of colonic microbiota among different treatment groups at 65 (d), 95 (e), and 125 (f) days old at level 3 using LEfSe analysis. C, control group; SOA, antibiotic supplementation in sow-offspring diets; SOP, probiotic supplementation in sow-offspring diets; SOS, synbiotic supplementation in sow-offspring diets. The replicates per group at 65 days old were 8. The replicates of the C, SOA, SOP, and SOS groups at 95 days old were 8, 8, 8, and 7, respectively. The replicates of the C, SOA, SOP, and SOS groups at 125 days old were 8, 5, 6, and 6, respectively.

the chloroalkane and chloroalkene degradation in the SOA group and selenocompound metabolism in the SOS group (Fig. 5c).

## Changes in colonic metabolome profiles of offspring piglets

The PCA combined with OPLS-DA was used to investigate the differences in colonic metabolites among different treatment groups (Fig. S4; Fig. 6). The metabolite composition of offspring piglets in the positive and negative models was clearly distinguished between the C and three treatment groups at 65, 95, and 125 days old (Fig. S4). OPLS-DA results indicated that metabolic profiles were distinct among the four groups at 65, 95, and 125 days old (Fig. 6). Moreover, a 200-cycle permutation test was used to identify the best-fitted OPLS-DA model (Fig. S5). These findings indicated that probiotic and synbiotic supplementation to sow's and offspring's diets could affect the colonic metabolite composition in offspring piglets.

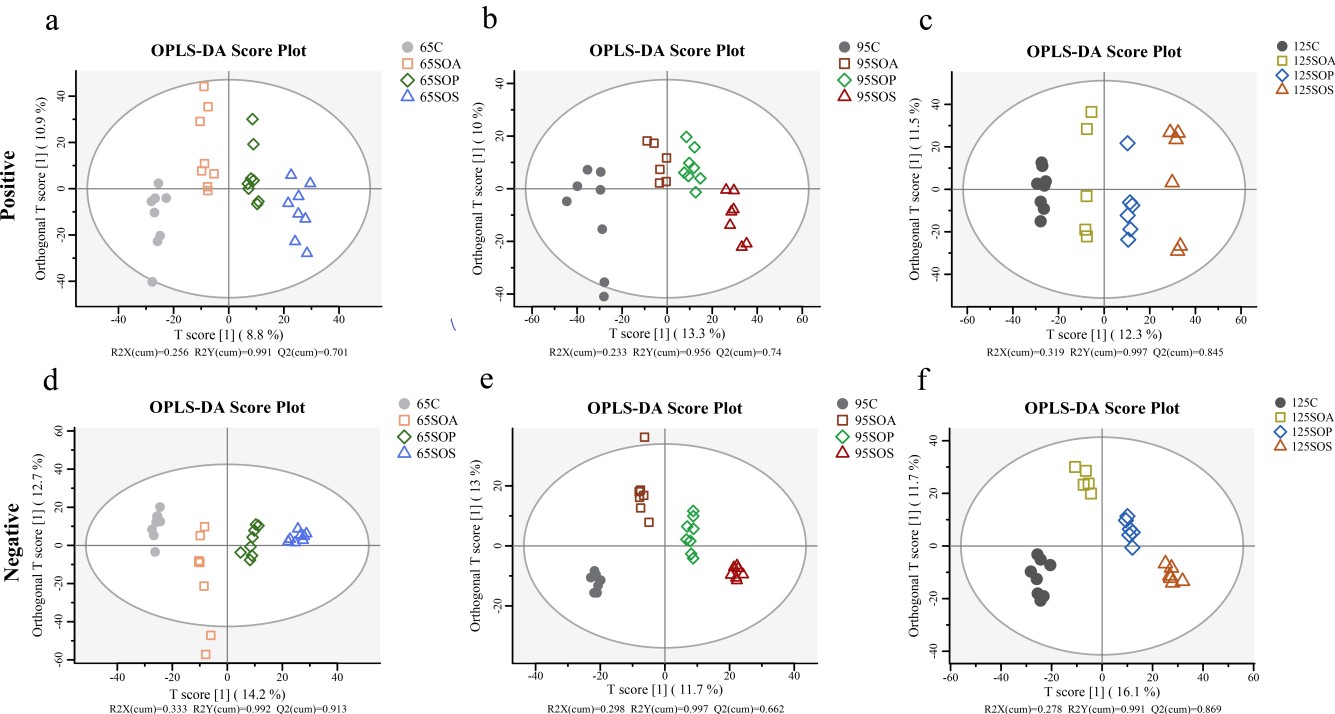

**FIG 6** OPLS-DA score plots based on the colonic metabolite in positive (a–c) and negative (d–f) ion modes at 65, 95, and 125 days old among different treatment groups. C, control group; SOA, antibiotic supplementation in sow-offspring diets; SOP, probiotic supplementation in sow-offspring diets; SOS, synbiotic supplementation in sow-offspring diets. The replicates per group at 65 days old were 8. The replicates of the C, SOA, SOP, and SOS groups at 95 days old were 8, 8, 8, and 7, respectively. The replicates of the C, SOA, SOP, and SOS groups at 125 days old were 8, 5, 6, and 6, respectively.

A total of 182 differential metabolites were identified among the four groups by the OPLS-DA model predictive value (VIP > 1) combined with $P < 0.05$. The differential analysis among the four groups showed that there were 46, 44, and 45 metabolites significantly changed at 65, 95, and 125 days old, respectively. In addition, the classification and analysis of differential metabolites revealed that these metabolites were mainly belonging to amino acids, carbohydrates, and lipids. The proportions of these three metabolites were 36.96%, 17.39%, and 23.91% at 65 days old (Fig. S6a); 29.55%, 11.36%, and 38.64% at 95 days old (Fig. S6b); and 31.11%, 8.89%, and 40.00% at 125 days old, respectively (Fig. S6c). The heatmap of differential metabolite analysis in the C, SOA, SOP, and SOS groups is presented in Fig. 7. At the VIP ≥ 2, there were five (including phenylacetaldehyde, beta-sitosterol, inosine, trioxilin A3, and L-serine; Fig. 7a), two (including guanidoacetic acid and pimelate; Fig. 7b), and seven (including putrescine, allopregnanolone, inosine, N-acetylhistamine, prostaglandin E2, 1-aminocyclopropanecarboxylic acid, and tetracosanoic acid; Fig. 7c) differential metabolites at 65, 95, and 125 days old, respectively.

A pairwise comparison between groups was carried out to further identify the core metabolites. There were 21, 38, and 25 common metabolites at 65, 95, and 125 days old, respectively (Fig. S6d through f). In addition, there were 8, 7, and 10 metabolites identified after combining these results with the pairwise comparison and ANOVA analysis at 65 (Fig. 8), 95 (Fig. 9), and 125 (Fig. 10) days old, respectively.

At 65 days old, in comparison with the C group, the normalized intensity of (+)-7-isojasmonic acid, N6,N6,N6-trimethyl-L-lysine, and palmitic acid were increased, whereas inosine, fructose 6-phosphate, and niacinamide were reduced in the SOA, SOP, and SOS groups ($P < 0.05$). In addition, the normalized intensity of cellobiose was the highest in the SOS group compared with the other three groups ($P < 0.05$). The intensity of isonicotinic acid in the SOS group was higher than that in the SOA group ($P < 0.05$). Moreover, the normalized intensity of palmitic acid was higher ($P < 0.05$), whereas

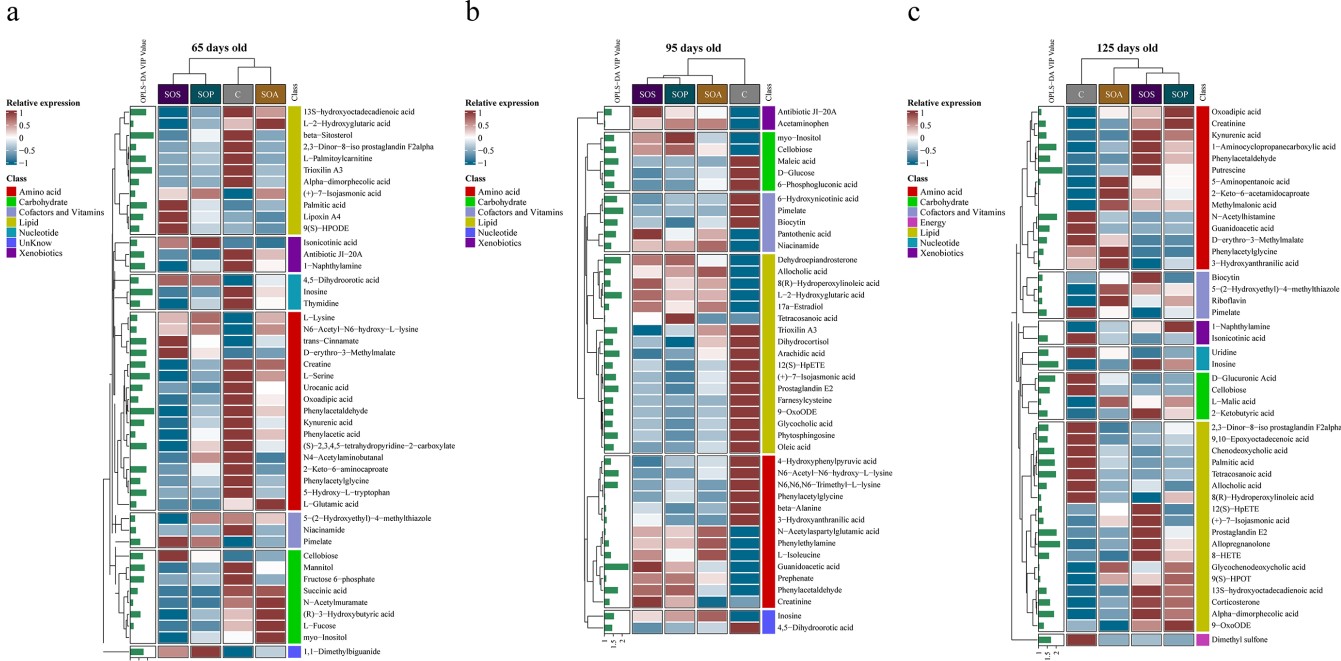

**FIG 7** The heatmap of differential metabolite analysis at 65, 95, and 125 days old (a–c). C, control group; SOA, antibiotic supplementation in sow-offspring diets; SOP, probiotic supplementation in sow-offspring diets; SOS, synbiotic supplementation in sow-offspring diets. The replicates per group at 65 days old were 8. The replicates of the C, SOA, SOP, and SOS groups at 95 days old were 8, 8, 8, and 7, respectively. The replicates of the C, SOA, SOP, and SOS groups at 125 days old were 8, 5, 6, and 6, respectively.

inosine was lower ($P < 0.05$) in the SOP and SOS groups, when compared with the C and SOA groups (Fig. 8).

As displayed in Fig. 9, at 95 days old, the normalized intensity of guanidoacetic acid, cellobiose, allocholic acid, niacinamide, and inosine were increased ($P < 0.05$), whereas N6-acetyl-N6-hydroxy-L-lysine and (+)-7-isojasmonic acid were decreased in the SOA, SOP, and SOS groups, when compared with the C group ($P < 0.05$). The normalized

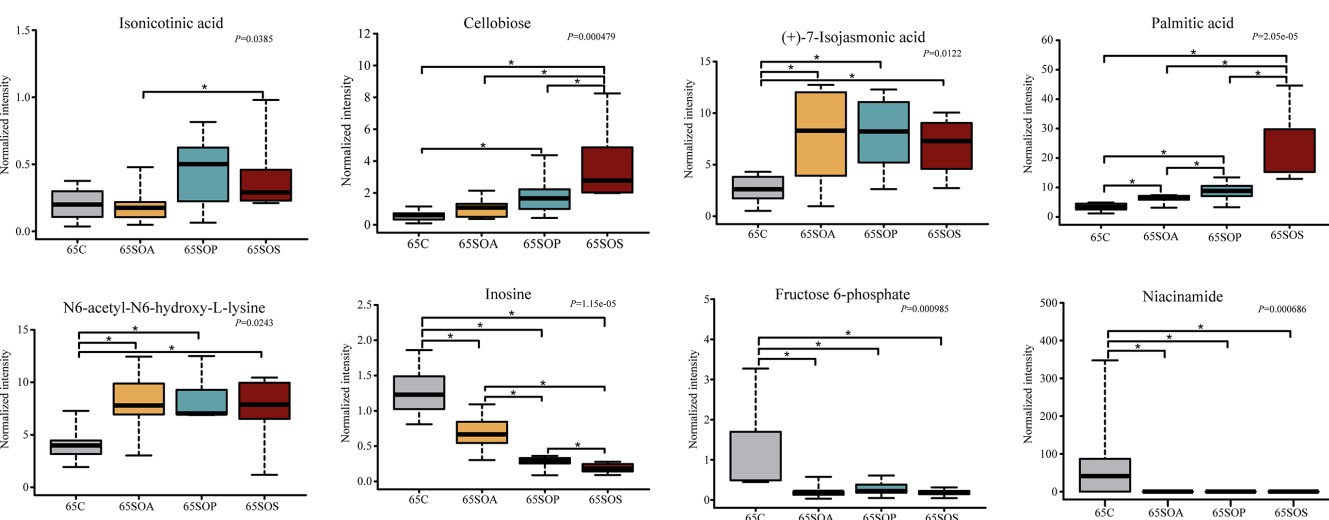

**FIG 8** The box map of differential metabolites at 65 days old. * $P < 0.05$. C, control group; SOA, antibiotic supplementation in sow-offspring diets; SOP, probiotic supplementation in sow-offspring diets; SOS, synbiotic supplementation in sow-offspring diets. The replicates per group at 65 days old were 8. The replicates of the C, SOA, SOP, and SOS groups at 95 days old were 8, 8, 8, and 7, respectively. The replicates of the C, SOA, SOP, and SOS groups at 125 days old were 8, 5, 6, and 6, respectively.

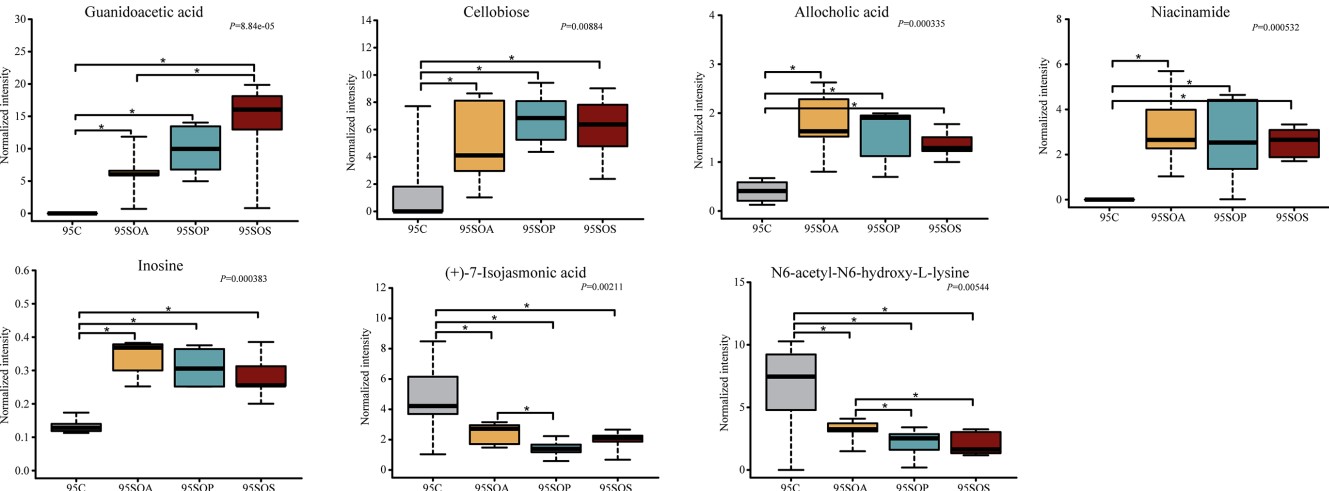

**FIG 9** The box map of differential metabolites at 95 days old. * P < 0.05. C, control group; SOA, antibiotic supplementation in sow-offspring diets; SOP, probiotic supplementation in sow-offspring diets; SOS, synbiotic supplementation in sow-offspring diets. The replicates per group at 65 days old were 8. The replicates of the C, SOA, SOP, and SOS groups at 95 days old were 8, 8, 8, and 7, respectively. The replicates of the C, SOA, SOP, and SOS groups at 125 days old were 8, 5, 6, and 6, respectively.

intensity of guanidoacetic acid was enhanced in the SOS group, whereas (+)-7-isojasmonic acid was reduced in the SOP group in comparison with the SOA group (P < 0.05). Moreover, N6-acetyl-N6-hydroxy-L-lysine intensity was reduced in the SOP and SOS groups compared with the SOA group (P < 0.05).

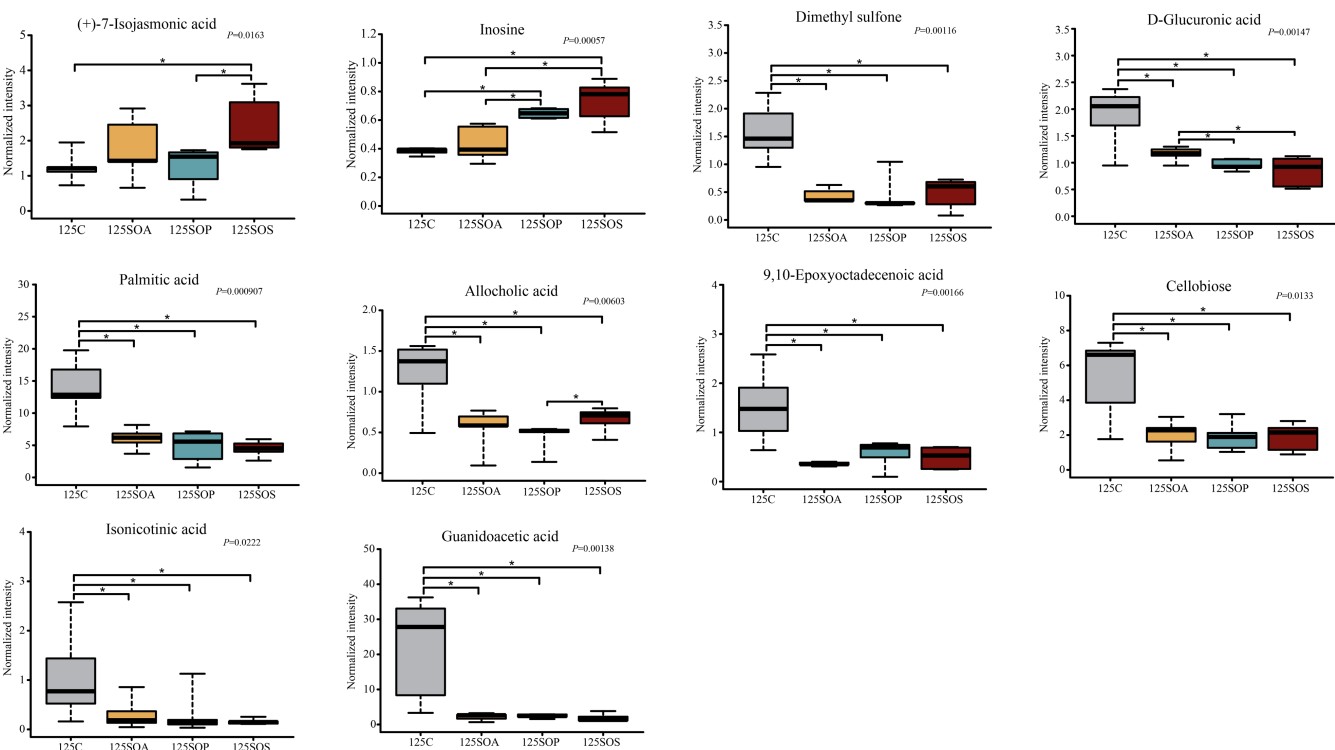

**FIG 10** The box map of differential metabolites at 125 days old. *, P < 0.05. C, control group; SOA, antibiotic supplementation in sow-offspring diets; SOP, probiotic supplementation in sow-offspring diets; SOS, synbiotic supplementation in sow-offspring diets. The replicates per group at 65 days old were 8. The replicates of the C, SOA, SOP, and SOS groups at 95 days old were 8, 8, 8, and 7, respectively. The replicates of the C, SOA, SOP, and SOS groups at 125 days old were 8, 5, 6, and 6, respectively.

As shown in Fig. 10, at 125 days old, the SOP and SOS groups had higher normalized intensity of inosine, whereas they had lower dimethyl sulfone, D-glucuronic acid, palmitic acid, allocholic acid, 9,10-epoxyoctadecenoic acid, cellobiose, isonicotinic acid, and guanidoacetic acid than the C group ($P < 0.05$). In addition, the SOP and SOS groups had higher normalized intensity of inosine, whereas they had lower D-glucuronic acid than the SOA group ($P < 0.05$). The (+)-7-isojasmonic acid-normalized intensity was increased in the SOS group in comparison with the C and SOP groups ($P < 0.05$). Furthermore, the correlations among the metabolites at three different ages were in line with the changes in the normalized intensity of these metabolites (Fig. 8 to 10; Fig. S7).

The metabolic pathways associated with these differential metabolites are shown in Fig. 11. Probiotic and synbiotic supplementation to sow's and offspring's diets influenced several metabolic pathways, including PPAR signaling pathway, GABAergic synapse, lysine degradation, butanoate metabolism, AMPK signaling pathway, and phenylalanine metabolism at 65 days old (Fig. 11a); non-alcoholic fatty liver disease, human papillomavirus infection, insulin signaling pathway, rheumatoid arthritis, and nicotinate and nicotinamide metabolism at 95 days old (Fig. 11b); and PPAR signaling pathway, human papillomavirus infection, rheumatoid arthritis, arginine and proline metabolism, and linoleic acid metabolism at 125 days old (Fig. 11c).

## Changes in colonic indole, skatole, bioamine, and SCFAs levels of offspring piglets

The changes in colonic microbial metabolites, including indole, skatole, bioamines, and SCFAs, are shown in Fig. 12a. At 65 days old, the skatole level was lower in the SOA and SOS groups compared with the C group ($P < 0.05$). At 95 days old, the indole and skatole levels were higher in the SOP group when compared with the C group ($P < 0.05$). At 125 days old, the indole level was increased in the SOA group, while the skatole level was reduced in the SOA and SOS groups, when compared with the C group ($P < 0.05$).

As presented in Fig. 12b through d, compared with that in the C group, the propionate level was higher in the SOA group, whereas valerate and isobutyrate levels in the SOP group, as well as the isovalerate level, were lower in the SOA and SOP groups at 65 days old ($P < 0.05$). At 95 days old, the butyrate level in the SOA, SOP, and SOS groups, as well as the isovalerate level, were reduced in the SOP group, whereas the valerate level was enhanced in the SOA group in comparison with the C group ($P < 0.05$). The SOA group had a higher acetate level than the C group at 125 days old ($P < 0.05$).

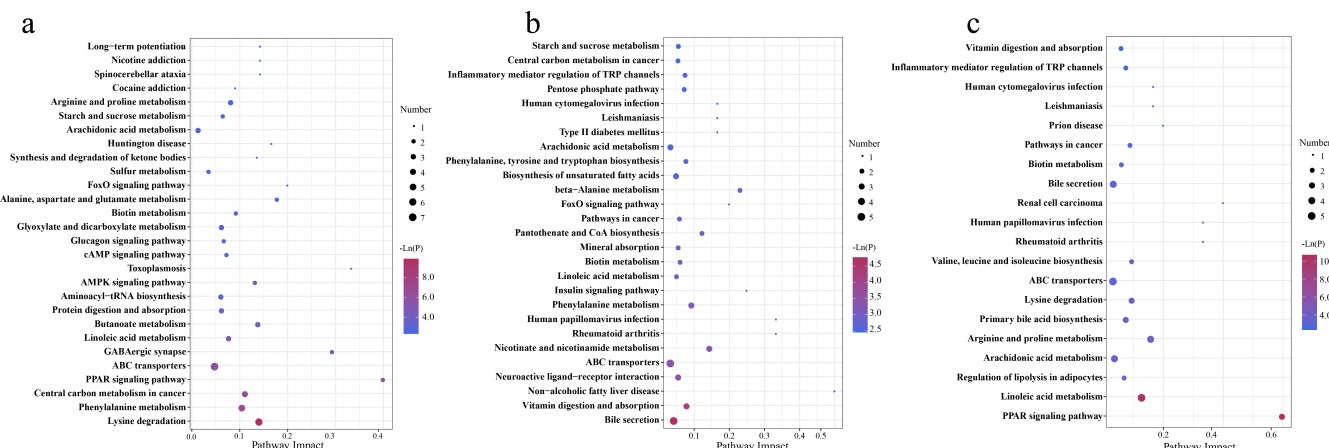

**FIG 11** The bubble chart of metabolic pathway enrichment analysis at 65 (a), 95 (b), and 125 (c) days old. The manipulated metabolic pathways are based on the analysis of differential colonic metabolites of offspring piglets among different treatment groups following the KEGG pathway database. The metabolome view shows all matched pathways according to the $P$ values from the pathway enrichment analysis and impact values from the topology analysis. The node colors varied from blue to maroon, indicating that the metabolites have different levels of significance.

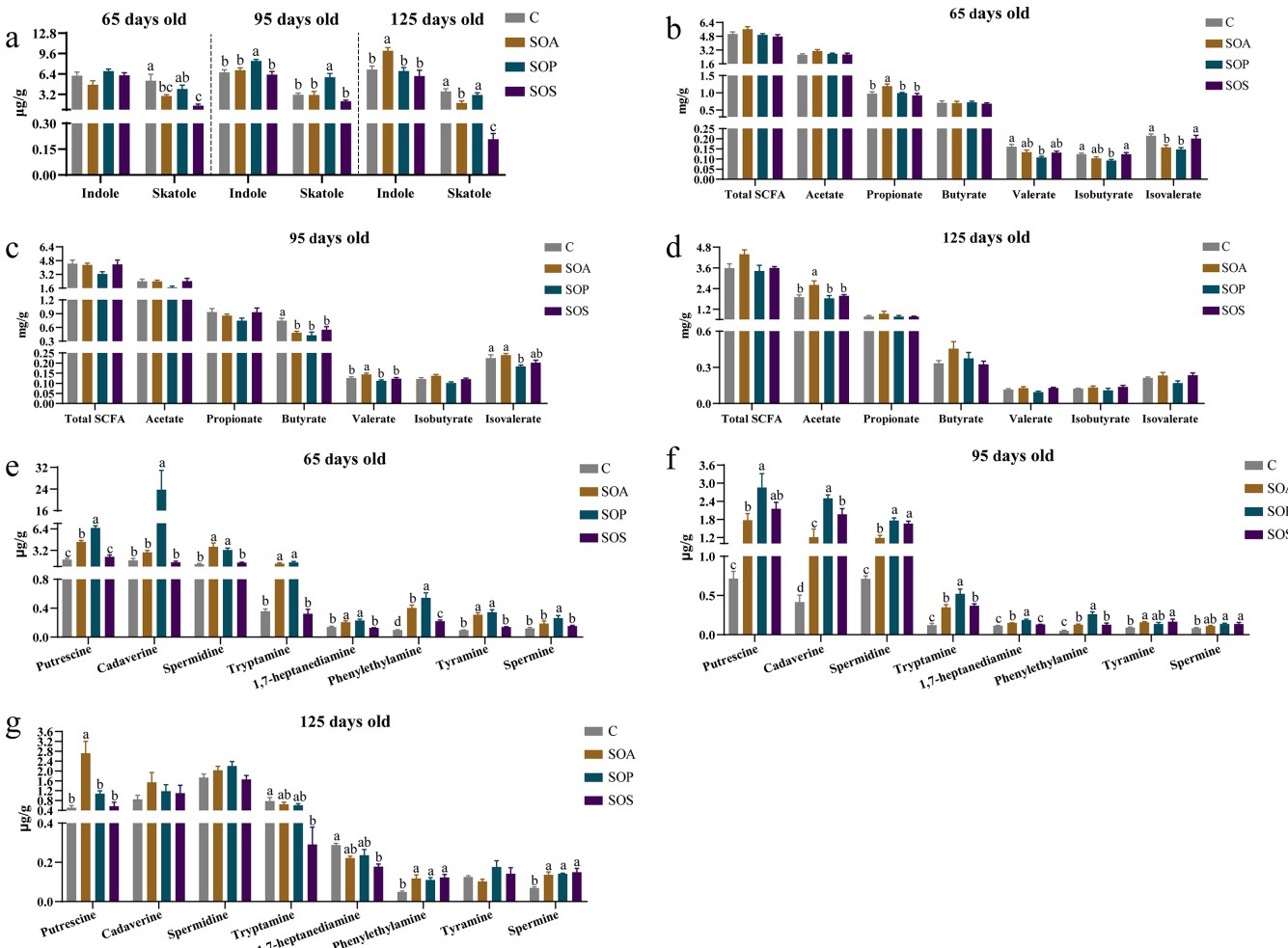

**FIG 12** Colonic concentrations of indole, skatole, SCFAs, and bioamines of offspring piglets at 65, 95, and 125 days old. Different lowercase letter mean a significant difference ($P < 0.05$). (a) Concentrations of indole and skatole; (b–d) concentrations of SCFAs; (f–g) concentrations of bioamines. C, control group; SOA, antibiotic supplementation in sow-offspring diets; SOP, probiotic supplementation in sow-offspring diets; SOS, synbiotic supplementation in sow-offspring diets. The replicates per group at 65 days old were 8. The replicates of the C, SOA, SOP, and SOS groups at 95 days old were 8, 8, 8, and 7, respectively. The replicates of the C, SOA, SOP, and SOS groups at 125 days old were 8, 5, 6, and 6, respectively.

As presented in Fig. 12e through g, at 65 days old, the levels of phenylethylamine of the SOA, SOP, and SOS groups; putrescine, spermidine, tryptamine, 1,7-heptanediamine, and tyramine of the SOA and SOP groups; and cadaverine and spermine of the SOP group were higher when compared with those of the C group ($P < 0.05$). At 95 days old, putrescine, cadaverine, spermidine, tryptamine, and phenylethylamine levels of the SOA, SOP, and SOS groups; 1,7-heptanediamine in the SOA and SOP groups; tyramine of the SOA and SOS groups; and spermine of the SOP and SOS groups were higher ($P < 0.05$) compared with the C group. At 125 days old, the levels of phenylethylamine and spermine of the SOA, SOP, and SOS groups and putrescine of the SOA group were higher, while tryptamine and 1,7-heptanediamine of the SOS group were lower than those in the C group ($P < 0.05$).

## Correlation analysis

There were a few correlations between the metabolites from metabolome analysis and colonic microbiota. At 65 days old, the cellobiose was positively correlated ($P < 0.05$) with *Treponema*, *Prevotella*, *L7A_E11*, *Sphaerochaeta*, and *RFN20* and inosine with *02d06* and *unclassified_Firmicutes*. Furthermore, the cellobiose was negatively correlated

(*P* < 0.05) with *unclassified_Firmicutes* and inosine with *unclassified_Ruminococcaceae*, *Parabacteroides*, *Lachnospira*, *Sphingomonas*, and *RFN20* (Fig. 13a). At 95 days old, the cellobiose was positively correlated (*P* < 0.05) with *p-75-a5* and *02d06*; guanidoacetic acid with *p-75-a5* and *Weissella*; and inosine with *Lactobacillus*, *Peptococcus*, and *Campylobacter*. Moreover, inosine was negatively correlated (*P* < 0.05) with *unclassified_Lachnospiraceae* and *Phascolarctobacterium* (Fig. 13b). At 125 days old, the guanidoacetic acid was positively correlated (*P* < 0.05) with *Clostridium*, *unclassified_Ruminococcaceae*, and *Lachnospira* and palmitic acid with *Clostridium*. Additionally, the normalized intensity of inosine was negatively correlated (*P* < 0.05) with *Clostridium* and *Lachnospira* (Fig. 13c).

The correlations between microbiota abundance with indole, skatole, SCFAs, and bioamines are presented in Fig. 13d through f. At 65 days old, *Faecalibacterium* was positively correlated (*P* < 0.05) with skatole, phenylethylamine, putrescine, cadaverine,

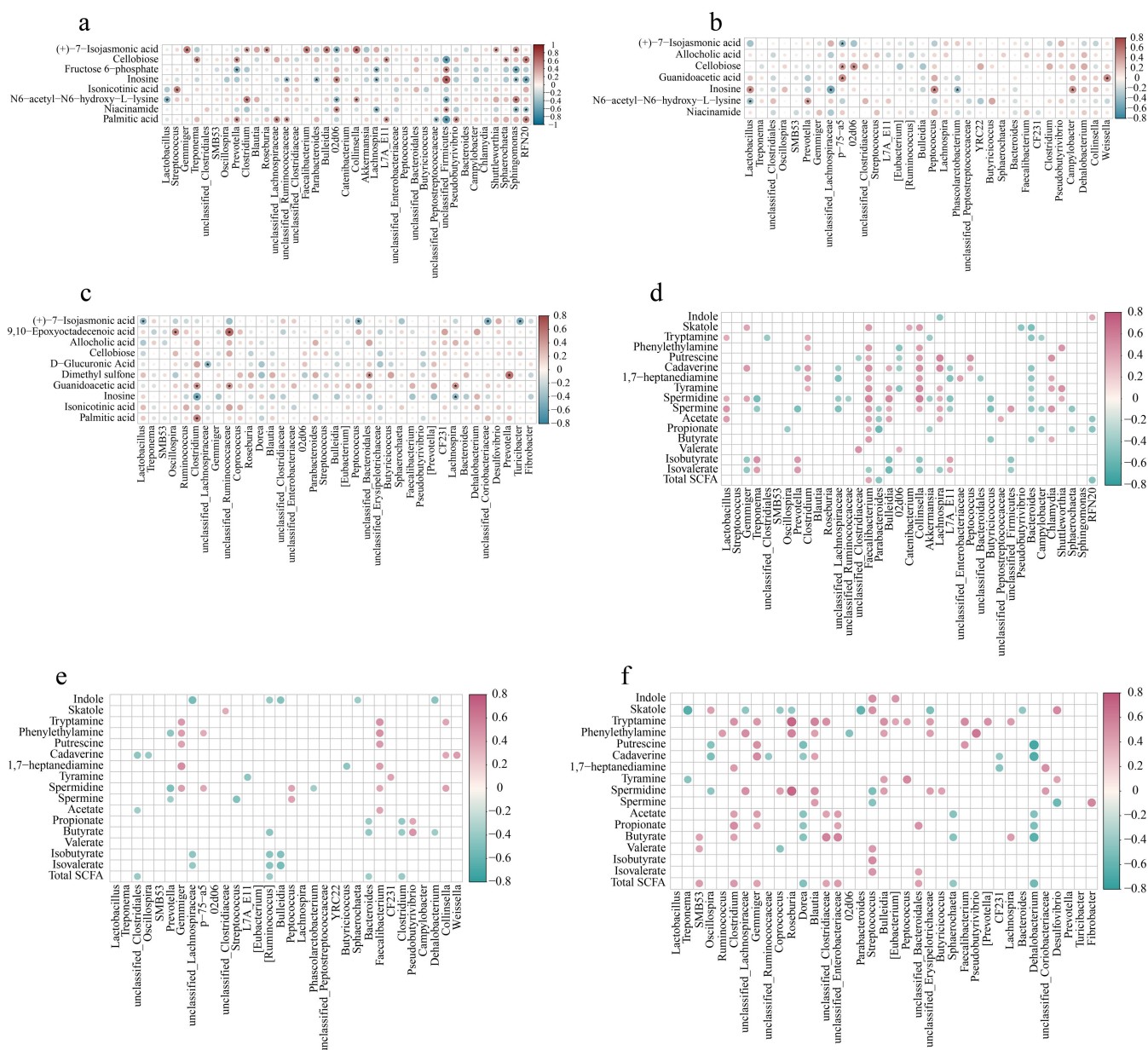

**FIG 13** Spearman correlation analysis of differential microbiota and metabolites in colonic contents of offspring piglets at 65, 95, and 125 days old. (a–c) The metabolites were from metabolomic analysis. (d–f) The metabolites were indole, skatole, SCFAs, and bioamines. The red represents positive correlation while the blue represents negative correlation. Asterisks (a–c) and red or blue circles (d–f) represent *P* < 0.05.

1,7-heptanediamine, tyramine, spermidine, spermine, acetate, propionate, butyrate, and total SCFA; *Lactobacillus* with tryptamine, spermidine, spermine, and acetate; and *Collinsella* with skatole, phenylethylamine, putrescine, cadaverine, 1,7-heptanediamine, tyramine, spermidine, and spermine. Furthermore, *Faecalibacterium* was negatively correlated ($P < 0.05$) with isovalerate and *Collinsella* with isobutyrate and isovalerate (Fig. 13d). At 95 days old, *Faecalibacterium* was positively correlated ($P < 0.05$) with tryptamine, phenylethylamine, putrescine, 1,7-heptanediamine, spermidine, and acetate and *Collinsella* with phenylethylamine, cadaverine, and spermidine (Fig. 13e). At 125 days old, *Faecalibacterium* was positively correlated ($P < 0.05$) with tryptamine and putrescine and *Clostridium* and *Gemmiger* with tryptamine, acetate, propionate, and total SCFA (Fig. 13f).

The correlations between appearance indexes and differential colonic microbiota and metabolites are shown in Fig. 14. At 65 days old, the content of C16:1 in LDM was positively correlated ($P < 0.05$) with colonic *Roseburia* and *unclassified_Enterobacteriaceae*; C16:1 in PMM with *Gemmiger*, *Blautia*, *Faecalibacterium*, *Collinsella*, and *unclassified_Firmicutes*; and plasma LDL-C with *Faecalibacterium*. Additionally, the content

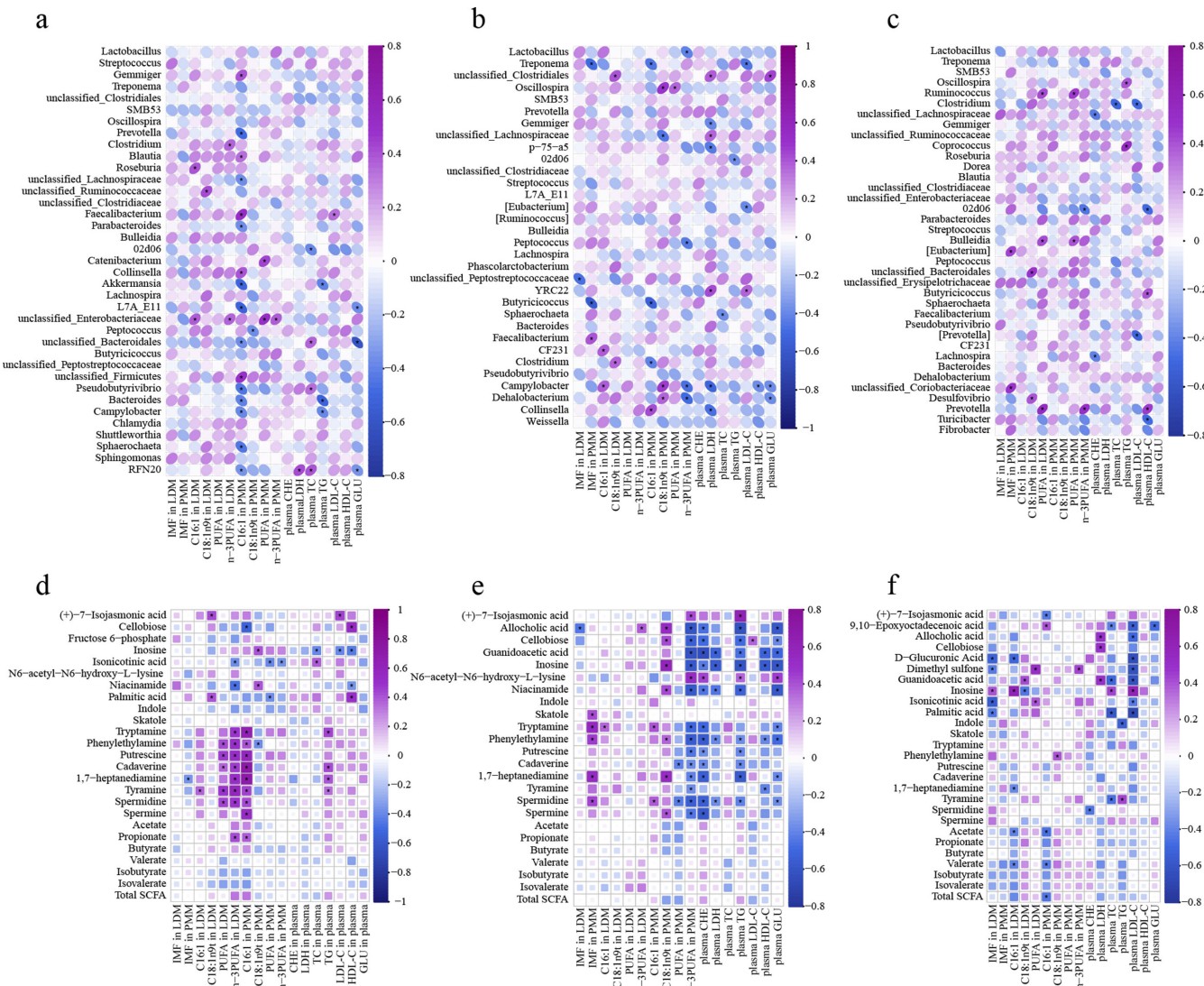

**FIG 14** Spearman correlation analysis between appearance index and differential microbiota (a–c) and metabolites (d–f) in colonic contents of offspring piglets at 65, 95, and 125 days old, respectively. Purple represents positive correlation while the midnight-blue represents negative correlation. Asterisk represents $P < 0.05$.

of C16:1 in PMM was negatively correlated ($P < 0.05$) with colonic *Prevotella, unclassified_Lachnospiraceae, Parabacteroides, Akkermansia, L7A_E11, unclassified_Bacteroidales, Pseudobutyrivibrio, Bacteroides, Campylobacter, Sphaerochaeta,* and *RFN20* (Fig. 14a). At 95 days old, the content of IMF in PMM was positively correlated ($P < 0.05$) with colonic *Faecalibacterium* and C16:1 in PMM with *Collinsella*. Moreover, the content of IMF in PMM was negatively correlated ($P < 0.05$) with *Butyricicoccus* and *Treponema*; C16:1 in PMM with *Treponema, Butyricoccus,* and *Clostridium*; and n-3 PUFA in PMM with *Campylobacter, Dehalobacterium,* and *Peptococcus* (Fig. 14b). At 125 days old, the content of IMF in PMM was positively correlated ($P < 0.05$) with colonic *[Eubacterium]* and *unclassified_Coriobacteriaceae*; n-3 PUFA in LDM with *Prevotella*; and plasma HDL-C with *Butyricicoccus* and *Prevotella*. Furthermore, the content of n-3 PUFA in LDM was negatively correlated ($P < 0.05$) with *02d06*; plasma LDL-C with *Clostridium* and *CF231*; and plasma HDL-C with *02d06* and *Turicibacter* (Fig. 14c).

At 65 days old, there were a few positive correlations between the meat quality indexes and lipid metabolism with the colonic metabolites from metabolome analysis. The contents of PUFA in LDM, n-3 PUFA in LDM, and C16:1 in PMM were positively correlated ($P < 0.05$) with colonic phenylethylamine, putrescine, cadaverine, tyramine, and spermidine (Fig. 14d). At 95 days old, the content of IMF in PMM was positively correlated ($P < 0.05$) with colonic skatole, tryptamine, phenylethylamine, and 1,7-heptanediamine. Additionally, n-3 PUFA in PMM and plasma CHE was negatively correlated ($P < 0.05$) with allocholic acid, cellobiose, guanidoacetic acid, inosine, tryptamine, phenylethylamine, putrescine, cadaverine, 1,7-heptanediamine, spermidine, and spermine (Fig. 14e). At 125 days old, there were a few correlations between the meat quality indexes and lipid metabolism with the colonic metabolites. The content of IMF in LDM was positively correlated ($P < 0.05$) with colonic inosine (Fig. 14f).

## DISCUSSION

Probiotics and synbiotics, common feed additives in livestock production, are widely used to enhance the animals' health and performance. Our previous studies demonstrated that diets supplemented with probiotics and synbiotics have beneficial effects on the reproductive performance and gut microbiota structure of sows, as well as the intestinal barrier function, antioxidant function, immunity function, and meat quality of offspring piglets (21–23). The present study determined the effects of supplementing probiotics and synbiotics in sow-offspring diets on colonic microbiota composition and their metabolites and the lipid metabolism of skeletal muscle of offspring piglets. The findings demonstrated that probiotics and synbiotics supplemented to sow's and offspring's diets have beneficial impacts on the IMF content, nutritional composition of skeletal muscles, colonic microbiota composition and their metabolites, and body metabolism, especially lipid metabolism of skeletal muscles.

The IMF content is the main index to evaluate meat quality related to meat color, tenderness, juiciness, and taste (24). Probiotic supplemented to sow's and offspring's diets reduced the IMF content in the PMM at 65 days old while increasing the IMF content in the PMM at 95 days old in our present study. This effect was related to the development of IMF, which mainly deposited after sexual maturity (25). In addition, at 125 days old, synbiotics supplemented to sow's and offspring's diets decreased the IMF content in the PMM. The specific mechanism needs further investigation.

Fatty acids not only reflect the nutritional value of meat but also have a great effect on cooked meat flavor (26). In addition, Grela et al. (27) revealed that dietary supplementing inulin and probiotics improved the fatty acid composition of pork. In the present study, probiotics and synbiotics supplemented to sow's and offspring's diets altered FA composition in muscle, suggesting that these two additives may play an important role in improving the nutritional value of meat by altering the FA composition. Chang et al. (28) demonstrated that dietary supplementation with *Lactobacillus plantarum* increased the levels of PUFA, C18:2n6c, and C18:3n3 in muscle. Our study also showed that probiotics supplemented to sow's and offspring's diets increased the levels

of PUFA and n-3 PUFA in the LDM, indicating that the nutritional value was improved in the LDM because n-3 PUFA is a healthy FA that reduces atherosclerosis and thrombosis (27). The C16:1 content is positively correlated with meat flavor (29). The current study found that probiotics supplemented to sow's and offspring's diets increased the C16:1 level in the LDM at 125 days old, demonstrating that meat flavor was improved. The C18:1n9t, a trans-fatty acid, has detrimental impacts on human health (30). In our present study, probiotic and synbiotic supplementation decreased the C18:1n9t level in the PMM at 65 days old. These findings demonstrated that probiotics and synbiotics supplemented to sow's and offspring's diets improved the nutritional value of the PMM. In addition, dietary antibiotic supplementation increased several beneficial FAs in the LDM and it also increased the C18:1n9t level in the PMM at 95 days old, indicating that dietary antibiotics have a favorable effect on improving the FA composition of LDM to some extent, while presenting the opposite effect on PMM.

The principal factors that affect fat accumulation include lipogenesis and lipolysis, and the expressions of genes related to lipid metabolism contributes to the meat quality in pigs (31). Among these genes, *PPARγ* plays an important role in fat accumulation by regulating the expressions of *LPL*, *FASN*, and *ACC* (31), whereas *FASN* and *ACC* promote fat deposition by regulating FA synthesis (13). Moreover, *SCD* is regulated by *PPARα* and participates in FA biosynthesis and composition (32), while *SREBP-1*, a lipogenic nuclear transcriptional regulator, directly affects the expressions of *ACC*, *FASN*, and *SCD* (33). Regarding lipid mobilization and FA oxidation, *ATGL* and *HSL* are known to be the major enzymes for TG catabolism in adipose tissue (34). Additionally, *CPT*-1 regulates mitochondrial FA oxidation and facilitates lipid mobilization (35). In the current study, probiotics and synbiotics supplemented to sow's and offspring's diets upregulated gene expressions related to lipogenesis (including *ACC*, *LPL*, *FASN*, *FABP4*, *PPARα*, *PPARγ*, *SCD*, and *SREBP-1*) and lipolysis (including *ATGL*, *HSL*, and *CPT*-1) in the LDM. These results could explain why dietary probiotic and synbiotic supplementation had no effect on the IMF content in the LDM. In addition, supplementing probiotics in sow-offspring diets resulted in upregulation of *ATGL*, *HSL*, and *CPT*-1 expressions in the PMM at 65 days old, while synbiotics resulted in upregulation of *CPT*-1 expression in the PMM at 125 days old, which may be the reason why the IMF content was decreased in the SOP group at 65 days old and SOS group at 125 days old in the current study. Furthermore, the upregulation of *PPARα* expression in the PMM of the SOP group may be explained by the increased IMF content at 95 days old after probiotic supplementation to sow's and offspring's diets.

The plasma TC, TG, HDL-C, and LDL-C levels could reflect the rate of lipid utilization (36). Moreover, the increase in CHE activity could reflect an increased lipid metabolism (37). Our results showed that probiotics and synbiotics supplemented to sow's and offspring's diets enhanced the plasma levels of HDL-C at 65 days old, CHE at 95 days old, and LDL-C at 125 days old, while reducing TG at 95 days old. Additionally, dietary synbiotic supplementation increased the plasma levels of HDL-C at 65 days old and TC at 65 and 125 days old. These results indicate that probiotic and synbiotic supplementation could improve lipid metabolism. Li et al. (38) also demonstrated that dietary synbiotics (including XOS, yeast cell wall, *B. licheniformis*, *B. subtilis*, and *C. butyricum*) supplementation could increase the HDL-C and decrease the TG concentrations in the plasma of chickens. The plasma levels of LDH and GLU could reflect energy and glucose metabolism, respectively (14). The SOP and SOS groups had decreased plasma levels of LDH and GLU at 95 days old in the current study, indicating that probiotics and synbiotics supplemented to sow's and offspring's diets improved energy and glucose metabolism in the offspring piglets.

It is widely accepted that higher microbial diversity is beneficial for the health and productivity of animals. In the present study, the richness and diversity of colonic microbiota of offspring piglets were not affected by dietary probiotic and synbiotic supplementation. Similarly, Zhang et al. (39) also found that *Lactobacillus* administration had no effects on the cecal and colonic microbial α-diversity. The β-diversity analysis

showed that probiotics and synbiotics supplemented to sow's and offspring's diets altered the colonic microbial community at 65, 95, and 125 days old, suggesting that supplementing these additives in sow-offspring diets may have advantageous effects on pigs by changing the intestinal microbiota structure of offspring piglets.

Firmicutes, Bacteroidetes, Actinobacteria, Proteobacteria, Fusobacteria, and Verrucomicrobia are the most dominant phyla in the intestine of pigs with Firmicutes and Bacteroidetes representing as the major two phyla (>90%) (40). In the present study, Firmicutes and Bacteroidetes were identified as the most dominant phyla at three different ages. In addition, the SOS group had enriched abundances of Firmicutes at 95 days old and Verrucomicrobia at 125 days old in the LEfSe analysis, suggesting that synbiotics supplemented to sow's and offspring's diets enhanced the abundance of beneficial bacteria. Verrucomicrobia aids glucose homeostasis in the gut and is also essential for a healthy gut due to its anti-inflammatory properties (41). In our study, probiotics and synbiotics supplemented to sow's and offspring's diets could promote gut health, as evidenced by sow-offspring synbiotic supplementation increasing Verrucomicrobia abundance at 95 and 125 days old and probiotics increasing Actinobacteria abundance at 65 days old. Similarly, a previous study illustrated that dietary yeast supplementation could increase Actinobacteria abundance, which may result from its advantageous effects of maintaining intestinal microbiota homeostasis (42).

*Lactobacillus* spp. play a vital role in maintaining gut microbiota homeostasis. Furthermore, some *Lactobacillus* species could shape the gut microbiota composition through the production of antimicrobial bacteriocins (43). The *Lactobacillus* abundance was dominant at different age stages at the genus level in the current study, suggesting that dietary probiotic and synbiotic supplementation are beneficial to maintain intestinal homeostasis of offspring piglets. *Faecalibacterium* is associated with a healthy gut status and produces butyrate through fermentation (44). The *Faecalibacterium* abundance in the SOP group displayed an increasing trend at 65 days old, and probiotic supplementation increased the abundance of *Turicibacter* at 125 days old in the current study, which is beneficial to gut health (45). Moreover, *Pseudobutyrivibrio* was enriched in the SOS group at 65 days old, which can produce butyrate as its main metabolite (46). A previous study also showed that dietary probiotic supplementation enhanced the abundances of Firmicutes, *Faecalibacterium*, and *Prevotella* in the colonic mucosa (47). However, sow-offspring antibiotic supplementation decreased several beneficial bacteria in the present study, including Bacteroidetes, Actinobacteria, *Unclassified_Lachnospiraceae*, and *Prevotella*. These findings suggest that probiotics and synbiotics supplemented to sow's and offspring's diets increase several butyrate-producing bacteria, whereas supplementing antibiotics reduces several beneficial bacteria.

The PICRUSt2 was used to reveal metagenome functional changes from 16S rRNA amplicon sequencing data and then identified the functional metabolic alterations. Our findings demonstrated that most of the pathways were related to the metabolism of carbohydrates, amino acids, cofactors and vitamins, terpenoids and polyketides, other amino acids, lipids, and energy at different ages. It is well known that gut microbiota plays a crucial role in the metabolism of carbohydrates, amino acids, and lipids (48). In the current study, the SOS group had enriched sphingolipid metabolism pathway at 95 days old and selenocompound metabolism pathway at 125 days old, suggesting that synbiotic supplementation enhanced the metabolism of lipids and other amino acids. Similarly, colonic metabolomics pathway enrichment analysis also showed that probiotics and synbiotics supplemented to sow's and offspring's diets affects the metabolism of carbohydrates, amino acids, and lipids. In addition, gut microbiota influences energy metabolism, immune reactions, intestine barrier functions, and adipogenesis by regulating several signaling pathways and interacting with immune and endocrine systems (49). In thepresent study, the PPAR, AMPK, and insulin signaling pathways were altered through dietary probiotic and synbiotic supplementation, suggesting that dietary probiotics and synbiotics could affect the physiology function and metabolism of the host by altering metabolic pathways.

In the present study, antibiotic supplementation enhanced xenobiotic biodegradation and metabolism at 125 days old, suggesting that dietary antibiotics may promote xenobiotic metabolism. This may result from the negative effects of antibiotics on the host. A previous study demonstrated that antibiotic treatment could affect the intestinal microbiota composition of animals involved in xenobiotic metabolism (50). In addition, maternal antibiotic supplementation alters the maternal microbiome and then influences gut microbiota homeostasis in offspring (51). The impairment of gut microbiome development and alteration of metabolic regulation later in life resulted from antibiotic perturbation of the piglet gut microbiota in early life (52); moreover, antibiotic exposure in finishing pigs also contributed to gut microbiota imbalance (53). However, the underlying mechanism warrants further investigation.

The gut microbiota could produce a wide range of microbial metabolites, which play a vital role in the host's physiological function and metabolic activities (54). In addition, colonic metabolites resulted from the combined action of gut microbiota and the host nutrient metabolism. The analysis of PCA and OPLS-DA in the current study indicated that probiotics and synbiotics supplemented to sow's and offspring's diets resulted in a clear separation of metabolite profiles. Furthermore, the heatmap analysis of metabolites also showed an obvious difference at different ages. These results indicate that probiotics and synbiotics supplemented to sow's and offspring's diets altered the metabolome profiles of offspring piglets. In addition, the differential metabolites mainly included amino acids, carbohydrates, and lipids. This is mainly due to the gut microbiota function that influences these nutrients' metabolism (55). When the VIP > 2, the differential metabolites were phenylacetaldehyde, beta-sitosterol, inosine, trioxilin A3, and L-serine at 65 days old, guanidoacetic acid and pimelate at 95 days old, and putrescine, allopregnanolone, inosine, N-acetylhistamine, prostaglandin E2, 1-aminocyclopropanecarboxylic acid, and tetracosanoic acid at 125 days old. These alterations may play beneficial roles on the offspring piglets.

Previous studies illustrated that inosine is an effective antioxidant and plays an important role in the maintenance of intestinal homeostasis (56); beta-sitosterol could exert anti-inflammatory activity in intestinal endothelial cells (57); guanidoacetic acid could improve meat quality (58); and putrescine could promote intestinal epithelial cell proliferation (59). Consistent with these results, sow-offspring diets supplemented with probiotics and synbiotics enhanced guanidoacetic acid and inosine intensity at 95 and 125 days old. In addition, colonic putrescine was also increased at 65 and 95 days old due to probiotic and synbiotic supplementation to sow's and offspring's diets. Moreover, inosine was positively associated with *Lactobacillus*, *Peptococcus*, and *Campylobacter* at 95 days old, as well as the guanidoacetic acid with *Clostridium* and *Lachnospira* at 125 days old. Cellobiose is a potential prebiotic, which has a beneficial role in maintaining gut health (60). In the current study, cellobiose intensity was increased at 65 and 95 days old by probiotic and synbiotic supplementation. All of these findings demonstrate that the beneficial effects of dietary probiotics and synbiotics may be related to the alteration of the colonic metabolite profiles.

Indole and skatole are the main end-products of tryptophan metabolism by gut microbes (61). In the current study, sow-offspring probiotic supplementation enhanced colonic indole concentration at 95 days old, and synbiotic supplementation reduced skatole concentration at 65 and 125 days old, indicating that dietary probiotics and synbiotics have critical roles in maintaining the function of the gut barrier and gut epithelial cell. This may resulted from the beneficial effect of indole on maintaining colon barrier integrity, whereas skatole may cause dysfunction of gut epithelial cells (62).

The SCFAs are gut microbiota-produced fermentation products that play a pivotal role in host health (63). In addition, branched-chain fatty acids, including isobutyrate and isovalerate, may cause detrimental effects on intestinal health and microbial composition (60). In the present study, sow-offspring probiotic supplementation decreased the colonic concentrations of isobutyrate at 65 days old and isovalerate at 65 and 125 days old, indicating that dietary probiotics could reduce the production of harmful

metabolites. Although sow-offspring probiotic and synbiotic supplementation increased some butyrate-producing bacteria, the correlation between colonic microbiota and SCFAs and the other SCFAs were not affected by the probiotic and synbiotic supplementation. The underlying mechanism requires further investigation.

Bioamines are produced by intestinal microbiota metabolizing their corresponding amino acids (64). Moreover, polyamines, including spermidine, spermine, and their precursor putrescine, are essential for normal gut epithelial renewal and barrier function (65). Our results demonstrated that probiotics and synbiotics supplemented to sow's and offspring's diets have beneficial impacts on gut health and could reinforce the metabolism of amino acids. Moreover, the correlation analysis indicated that colonic *Faecalibacterium* was positively correlated with spermidine, spermine, and putrescine. This may be the reason why dietary probiotics and synbiotics enhance the production of polyamines. Therefore, polyamine modulation by gut microbiota and probiotics could be a good strategy to achieve beneficial effects for the host (66).

A previous study demonstrated that *Roseburia* spp. and *Clostridium* spp. produce conjugated linoleic acid and SCFAs, respectively, which can effectively improve the pork quality by reducing body fat (67). Moreover, *L. plantarum ZJ316* can inhibit the growth of opportunistic pathogens and increase the villus height of intestine through its metabolites, thus improving the growth performance and meat quality of pigs (68). In the current study, the appearance indexes were correlated with colonic microbiota and metabolites, and the positive correlation included between C16:1 in PMM with *Gemmiger*, *Blautia*, *Faecalibacterium*, and *Collinsella*; IMF in PMM and plasma LDL-C with *Faecalibacterium*; and n-3 PUFA in PMM with *Dehalobacterium*. Similarly, a previous study demonstrated that *Blautia* and *Faecalibacterium* are correlated with lipid metabolism (69). Moreover, *Blautia* may inhibit fat accumulation in adipocytes, in turn promoting lipid metabolism in other tissues (70). *Collinsella* is positively correlated with dietary protein intake, body weight, and fat content (71). *Dehalobacterium* is associated with increased blood lipid and body mass index, which is beneficial to maintaining body metabolism and health (72). These findings imply that the improvement of meat quality and lipid metabolism might be related to the alterations of gut microbiota and metabolites.

The gut microbiota metabolites are also associated with meat quality and body metabolism. The SCFAs can be rapidly absorbed and utilized by the host, affecting lipid metabolism and muscular IMF content (73). However, there were a few positive correlations between the meat quality indexes and lipid metabolism with the colonic SCFAs in the present study. This may have resulted from the lack of significant changes in colonic SCFAs content.

Bioamines are involved in several processes related to transcription, translation, growth, metabolism, and other functions (20). However, the effect of bioamines on meat quality has not been reported. The present study showed several positive correlations between colonic bioamines with meat quality indexes and lipid metabolism at 65 and 95 days old. The specific physiological significance and mechanism require further investigation. A previous study demonstrated that spermidine is a key factor in the process of adipogenesis (74). In the current study, spermidine was positively correlated with the IMF in PMM, whereas it was negatively correlated with plasma TG at 95 days old, which could be explained by the fact that probiotics and synbiotics supplemented to sow's and offspring's diets enhanced the IMF in the PMM and improved the lipid metabolism.

## Conclusions

In summary, the integral sow-offspring probiotic and synbiotic supplementation improves the fatty acid composition of skeletal muscle to some extent, as well as the metabolism of lipid, amino acid, and glucose of offspring piglets, and increases the colonic beneficial bacteria (including Actinobacteria, Firmicutes, Verrucomicrobia, *Faecalibacterium*, *Pseudobutyrivibrio*, and *Turicibacter*) and increases the colonic

metabolome profiles, such as guanidoacetic acid, inosine, cellobiose, indole, and polyamine. These alterations may have beneficial effects on the nutritional values of meat, gut health, and body metabolism. However, antibiotic supplementation in sow-offspring diets decreases several beneficial bacteria (including Bacteroidetes, Actinobacteria, *Unclassified_Lachnospiraceae*, and *Prevotella*). In addition, antibiotic supplementation has a favorable effect on improving the fatty acid composition of LDM to some extent, whereas it presented the opposite effect on the PMM. These findings provide new insights into the application of probiotics and synbiotics on sow-offspring diets for improving the meat quality by reshaping the gut/intestinal microbiota and body metabolism of offspring pigs.

## ACKNOWLEDGMENTS

The present study was jointly supported by the National Key Research and Development Project (grant number 2023YFD1301305), Key Project of Regional Innovation and Development Joint Fund of National Natural Science Foundation of China (U20A2056), Open Fund of Key Laboratory of Agro-ecological Processes in Subtropical Region, Chinese Academy of Sciences (ISA2023203), Hunan Natural Science Foundation (2022JJ30643), and Special Funds for Construction of Innovative Provinces in Hunan Province (2019RS3022).

We thank all staff and postgraduate students of Hunan Provincial Key Laboratory of Animal Nutritional Physiology and Metabolic Process for animal feeding and sample collection and technicians from CAS Key Laboratory of Agro-ecological Processes in Subtropical Region for providing technical assistance.

X.K. and Y.Y. conceived the experiment. Q.Z. carried out the feeding experiment and sample analysis, organized and analyzed all the data, and wrote the manuscript. C.L., R.L., and Y.L. assist in the completion of sample collection. Q.Z., X.K., and M.A.K.A. revised the manuscript. All authors reviewed and approved the final manuscript.

## AUTHOR AFFILIATIONS

[1]Key Laboratory of Agro-ecological Processes in Subtropical Region, Hunan Provincial Key Laboratory of Animal Nutritional Physiology and Metabolic Process, National Engineering Laboratory for Pollution Control and Waste Utilization in Livestock and Poultry Production, Institute of Subtropical Agriculture, Chinese Academy of Sciences, Changsha, Hunan, China
[2]College of Advanced Agricultural Sciences, University of Chinese Academy of Sciences, Beijing, China

## AUTHOR ORCIDs

Qian Zhu http://orcid.org/0000-0003-3808-4062
Md. Abul Kalam Azad http://orcid.org/0000-0003-0118-5708
Xiangfeng Kong http://orcid.org/0000-0001-8034-6682

## FUNDING

| Funder | Grant(s) | Author(s) |
| --- | --- | --- |
| National Key Research and Development Project | 2023YFD1301305 | Xiangfeng Kong |
| Key Project of Regional innovation and Development Joint Fund of National Natural Science Foundation of China | U20A2056 | Xiangfeng Kong |
| Open Fund of Key Laboratory of Agro-ecological Processes in Subtropical Region, Chinese Academy of Sciences | ISA2023203 | Qian Zhu |

| Funder | Grant(s) | Author(s) |
|---|---|---|
| Hunan Natural Science Foundation | 2022JJ30643 | Md. Abul Kalam Azad |
| Special Funds for Construction of Innovative Provinces in Hunan Province | 2019RS3022 | Xiangfeng Kong |

## DATA AVAILABILITY

The data generated or analyzed during the current study are available from the corresponding author by reasonable request.

## ETHICS APPROVAL

All animal protocols used in the current study were in accordance with the Chinese Guidelines for Animal Welfare and approved by the Animal Care and Use Committee of the Institute of Subtropical Agriculture, the Chinese Academy of Sciences (No. ISA-2018-071).

## ADDITIONAL FILES

The following material is available online.

### Supplemental Material

**Supplemental material (mSystems00048-24-s0001.docx).** Supplemental figures and tables.

### Open Peer Review

**PEER REVIEW HISTORY (review-history.pdf).** An accounting of the reviewer comments and feedback.

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
