## [Reviewer comments · mSystems]

Dietary probiotics and synbiotics supplementation starting from maternal gestation improves muscular lipid metabolism in offspring piglets by reshaping colonic microbiota and metabolites

Qian Zhu, Abul Azad, Ruixuan Li, Chenjian Li, Yang Liu, yulong yin, and Xiangfeng Kong

Corresponding Author(s): Xiangfeng Kong, Institute of Subtropical Agriculture, Chinese Academy of Sciences

Review Timeline:

Submission Date:	January 10, 2024
Editorial Decision:	February 26, 2024
Revision Received:	March 21, 2024
Editorial Decision:	April 15, 2024
Revision Received:	April 15, 2024
Accepted:	April 16, 2024

Editor: Juliette Hayer

Reviewer(s): Disclosure of reviewer identity is with reference to reviewer comments included in decision letter(s). The following individuals involved in review of your submission have agreed to reveal their identity: Yong Su (Reviewer #1); Xinxia Wang (Reviewer #2)

Transaction Report:

DOI: <https://doi.org/10.1128/msystems.00048-24>

Re: mSystems00048-24 (Dietary probiotics and synbiotics supplementation starting from maternal gestation improves muscular lipid metabolism in offspring piglets by reshaping colonic microbiota and metabolites)

Dear Prof. Xiangfeng Kong:

I am pleased to announce that your manuscript has been accepted for publication in mSystems, with minor modifications. Please take into account the reviewers valuable comments, I think the manuscript will benefit from addressing the questions they raised. Also, please make sure that the "Data availability" paragraph complies with ASM data policy, and contains the appropriate links to the public data.

Revision Guidelines

Sincerely,
Juliette Hayer
Editor
mSystems

Reviewer #1 (Comments for the Author):

The work entitled "Dietary probiotics and synbiotics supplementation starting from maternal gestation improves muscular lipid metabolism in offspring piglets by reshaping colonic microbiota and metabolites" by Zhu et al. provided valuable data about the application of feed additives (probiotics and synbiotics) subsequently to sows and their offspring piglets on the skeletal muscle lipid metabolism and gut microbiota and metabolites. The authors generated a lot of data to verify their hypothesis, and their findings showed that application of probiotics and synbiotics to sows and offspring is beneficial to improve the quality through modulating gut microbiota and lipid metabolism of offspring pigs. Generally, this manuscript was well-designed and carefully organized. However, there are still some minor concerns worthy of the attention of the authors:

Detailed comments:

1. What amounts of feed were provided daily? Restricted or ad libitum?
2. What about the sexual distribution of the experimental pigs? Male/female? Provide more details.
3. Was there any vaccination program for sows and offspring piglets during the trial?
4. What were the criteria for eight piglets for sampling (L118-119)?
5. Which ANOVA was used for statistical analysis? One/two way?
6. Keep the consistent format throughout the text ($P < 0.05$ or $P < 0.05$).
7. What other health indexes (L577)?

Minor points:

L6: with these>> of these

L16: as well as upregulated...

L25-26: different age stages.

L110-113: mL/day>> mL/d; keep consistent format throughout the text.

L123: "Sacrifice" is more likely religious word; replace with the word "euthanized".

L124: Location and weight of the colon contents?

L207/208: $P \leq 0.05$; $VIP \geq 1$.

L210: using the

L237: Define SOA, SOP, and SOS at their first appearance in the text.

L257: C18:1n9t content.

L278: Define MyoG at its first appearance in the text.

L306: Details of these 88 samples should be provided.

L 308: $P > 0.05$.

L411: $P < 0.05$.

L419: $VIP \geq 2$.

L564: IMF content.

L612-613:resulted in upregulation of ATGL, HSL, and CPT-1 expressions...

L615: Which specific groups had decreased IMF?

Tables: Please provide table note for "".

Table 2: C18:3n3 (95 day-old); Why comparison without detection values of other groups?

Reviewer #2 (Comments for the Author):

Probiotics and synbiotics, as common feed additives in livestock production, are widely used to enhance the animals' health and performance. The authors present an interesting question in this paper: Can dietary probiotics and synbiotics supplementation improve muscular lipid metabolism in offspring piglets by reshaping colonic microbiota and metabolites? There is a large amount of presented but not seated in any statements or data that would make the work meaningfully applicable to swine production. My detailed assessment as follows:

1. Please provide the information of the probiotics and synbiotics products used in the present study, e.g., purity, source, composition, etc.
2. Please change [12000] to [12,000], use commas when numbers exceed four digits. Please check and correct the full text.
3. What amounts of feed were provided daily? Restricted or ad libitum?
4. Feed intake and body weight must be provided?
5. Given the nature of the dietary intervention (probiotics and synbiotics supplementation) and the expectation of major changes in microbiota composition and metabolism, measuring small intestinal microbiota seems critical in this study. Why only colonic microbiota was measured?
6. Which ANOVA was used for statistical analysis? One/two way?
7. How about the growth performance of the offspring piglets after dietary supplementation? What about mortality rate during the experimental period from 35 to 125 days?
8. Line 100 "Sows were housed in individual pens (2.2 × 0.6 m) during gestation" in the pigging of the trail what is the number of sow month gestation?
9. How long did the sows receive feed with experimental supplements? Whole pregnancy or less? It needs to be added.
10. Lines 113-114: although the supplier, ingredient, and feeding method of these additives were consistent with the previous study (12), it is worth giving the name of the probiotic and prebiotic here.
11. What were the criteria for eight piglets for sampling (L118-119)?
12. What about the environmental conditions surrounding the sows and piglets? For example, season, temperature, relative

humidity, the nature of housing, and the form of feed provided?

13. Why is there a difference in the amounts of probiotics, synbiotics, and antibiotics during the different trial periods? If the addition is to the feed, then the difference in the amount of feed consumed between the sow and the offspring is variable, and not the difference in the concentration of the feed additives used.

14. Are there any data on pregnant sows during the first period of the experiment, such as the rate of feed consumption, average weight of the offspring pigs at birth, as well as their body weight at weaning and mortality rate during lactation period as affected by feed additives used?

15. Authors should re-format the references based on journal format. See the instructions for authors.

Response letter

Reviewer #1 (Comments for the Author):

The work entitled "Dietary probiotics and synbiotics supplementation starting from maternal gestation improves muscular lipid metabolism in offspring piglets by reshaping colonic microbiota and metabolites" by Zhu et al. provided valuable data about the application of feed additives (probiotics and synbiotics) subsequently to sows and their offspring piglets on the skeletal muscle lipid metabolism and gut microbiota and metabolites. The authors generated a lot of data to verify their hypothesis, and their findings showed that application of probiotics and synbiotics to sows and offspring is beneficial to improve the quality through modulating gut microbiota and lipid metabolism of offspring pigs. Generally, this manuscript was well-designed and carefully organized. However, there are still some minor concerns worthy of the attention of the authors:

Response: We would like to thank you for your valuable time and insightful comments and suggestions on our manuscript. The comments and suggestions are helpful to improve the quality of our manuscript. We have read all comments and suggestions carefully and made corrections accordingly.

Detailed comments:

1. What amounts of feed were provided daily? Restricted or *ad libitum*?

Response: The sows were fed with 0.8, 1.0, 1.2, 1.5, and 2.0 kg of pregnancy diets from days 1–15, 16–30, 31–75, 76–90, and 91–105 of pregnancy, respectively; fed with 1 kg of pregnancy feed diets a week before parturition and *ad libitum* access after three days of parturition; and fed with 2.4 kg of lactation diets until weaning. The offspring pigs had *ad libitum* to feed at all times.

We have added these details in the revised manuscript (L117-121 and L129-130).

2. What about the sexual distribution of the experimental pigs? Male/female? Provide more details.

Response: One male and one female piglet close to the average BW per litter were selected and transferred to the nursery house for the subsequent feeding trial. We have added this detail in the revised manuscript (L124-125).

3. Was there any vaccination program for sows and offspring piglets during the trial?

Response: Feeding and management (including vaccination program) for sows and offspring piglets were carried out according to the standard operations of commercial pig farms. We have added these in the revised manuscript (L143-415).

4. What were the criteria for eight piglets for sampling (L118-119)?

Response: Eight offspring pigs per group (one pig from each pen with an average BW of the pen) at each time point (65, 95, and 125 day-old) were selected for sampling. We have added these details in the revised manuscript (L148-149).

5. Which ANOVA was used for statistical analysis? One/two way?

Response: The one-way ANOVA was used for statistical analysis. We have added this in the revised manuscript (L254).

6. Keep the consistent format throughout the text ($P < 0.05$ or $P < 0.05$).

Response: We have checked and revised throughout the manuscript carefully.

7. What other health indexes (L577)?

Response: This sentence means that the changes in fatty acid composition could improve the fatty acid content related to the health-promoting index. To avoid ambiguity, we removed "and several indexes related to beneficial health" in the revised manuscript (L607-608).

Minor points:

L6: with these>> of these

Response: We have corrected it (L21).

L16: as well as upregulated...

Response: We have corrected it (L30).

L25-26: different age stages.

Response: Corrected (L40-41).

L110-113: mL/day>> mL/d; keep consistent format throughout the text.

Response: We have checked and revised these throughout the manuscript.

L123: "Sacrifice" is more likely religious word; replace with the word "euthanized".

Response: We have replaced the word "sacrifice" to "euthanized" (L153).

L124: Location and weight of the colon contents?

Response: Approximately 2 g of the colon contents (middle section) were collected into sterile centrifuge tubes. We have added this in the revised manuscript (L154).

L207/208: $P \leq 0.05$; $VIP \geq 1$.

Response: We have corrected it in the revised manuscript (L237/238).

L210: using the

Response: We have added "the" in the revised manuscript (L240).

L237: Define SOA, SOP, and SOS at their first appearance in the text.

Response: We have added the details about SOA, SOP, and SOS at their first appearance in the text.

L257: C18:1n9t content.

Response: We have revised it in the revised manuscript (L289).

L278: Define MyoG at its first appearance in the text.

Response: There is an error in this place, we have checked the results and revised the results and discussion section.

L306: Details of these 88 samples should be provided.

Response: These 88 samples included 32, 31, and 25 samples at 65, 95, and 125 day-old. We have added this information in the revised manuscript (L337-338).

L 308: $P > 0.05$.

Response: We have corrected it (L341).

L411: $P < 0.05$.

Response: Corrected (L443).

L419: $VIP \{ \text{greater than or equal to} \} 2$.

Response: We have revised it in the revised manuscript (L451).

L564: IMF content.

Response: We have added "content " in the revised manuscript (L596).

L612-613:resulted in upregulation of ATGL, HSL, and CPT-1 expressions...

Response: We have revised it in the revised manuscript (L640-641).

L615: Which specific groups had decreased IMF?

Response: The IMF content was decreased in the SOP group at 65 day-old and SOS group at 125 day-old. We have added this in the revised manuscript (L640-641).

Tables: Please provide table note for "-".

Response: We have provided table note for "-" in Tables 1 and 2.

Table 2: C18:3n3 (95 day-old); Why comparison without detection values of other groups?

Response: We have deleted the comparison without detection values.

Response letter

Reviewer #2 (Comments for the Author):

Probiotics and synbiotics, as common feed additives in livestock production, are widely used to enhance the animals' health and performance. The authors present an interesting question in this paper: Can dietary probiotics and synbiotics supplementation improve muscular lipid metabolism in offspring piglets by reshaping colonic microbiota and metabolites? There is a large amount of presented but not seated in any statements or data that would make the work meaningfully applicable to swine production. My detailed assessment as follows:

Response: Sincerest thanks for your valuable time and constructive comments on our manuscript. We have read all comments and suggestions carefully and made corrections accordingly.

1. Please provide the information of the probiotics and synbiotics products used in the present study, e.g., purity, source, composition, etc.

Response: The probiotics mixture was provided by Hunan Lifeng Biotechnology Co., Ltd. (Changsha, China) and contained *Lactobacillus plantarum* B90 (CGMCC1.12934) $\geq 1 \times 10^8$ CFU/mL and *Saccharomyces cerevisiae* P11 (CGMCC2.3854) $\geq 0.2 \times 10^8$ CFU/mL. The XOS ($\geq 35\%$) was provided by Shandong Longlive Biotechnology Co., Ltd. (Shandong, China) and contained xylobiose (55%), xylotriose (25%), xyloetraose (10%), xylopentose (5%), xylohexaose (3%), and xyloheptaose (2%), which met the feed additive of XOS recommended requirements (GB/T23747-2009). The information reported in our previous study (Zhu Q, Song M, Azad MAK, Cheng Y, Liu Y, Liu Y, Blachier F, Yin Y, Kong X. 2022. Probiotics or synbiotics addition to sows' diets alters colonic microbiome composition and metabolome profiles of offspring pigs. *Front Microbiol* 13:934890.). We also added this information in the revised manuscript (L134-140).

2. Please change [12000] to [12,000], use commas when numbers exceed four digits. Please check and correct the full text.

Response: We have changed [12000] to [12,000] and corrected throughout the full text.

3. What amounts of feed were provided daily? Restricted or ad libitum?

Response: The sows were fed with 0.8, 1.0, 1.2, 1.5, and 2.0 kg of pregnancy diets from days 1–15, 16–30, 31–75, 76–90, and 91–105 of pregnancy, respectively; fed with 1 kg of pregnancy diets a week before parturition and *ad libitum* after three days of parturition; and fed with 2.4 kg of lactation diets until weaning. The offspring pigs had *ad libitum* access to feed at all times.

We have added these details in the manuscript (L117-121 and L129-130).

4. Feed intake and body weight must be provided?

Response: The growth performance (including BW, ADG, ADFI, and F/G) of offspring pigs have been reported in our previous study (Zhu Q, Azad MAK, Dong H, Li C, Li R, Cheng Y, Liu Y, Yin Y, Kong X. 2023. Sow-offspring diets supplemented with probiotics and synbiotics are associated with offspring's growth performance and meat quality. *Int J Mol Sci* 24:7668). The specific data is as follows:

Items	C Group	SOA Group	SOP Group	SOS Group	SEM	P-values
BW, kg						
35 d-old	4.97	5.04	4.71	4.78	0.135	0.275
65 d-old	9.37	9.25	9.49	8.96	0.168	0.162
95 d-old	14.05	15.16	13.52	13.61	0.548	0.151
125 d-old	22.67 ^b	27.23 ^a	28.29 ^a	19.28 ^c	0.912	<0.001
ADG, kg/d						
35–65 d-old	0.15	0.15	0.15	0.14	0.006	0.435
66–95 d-old	0.18	0.17	0.16	0.17	0.012	0.693
96–125 d-old	0.27 ^b	0.30 ^b	0.37 ^a	0.18 ^c	0.023	<0.001
ADFI, kg/d						
35–65 d-old	0.41	0.43	0.42	0.44	0.009	0.312
66–95 d-old	0.64	0.65	0.69	0.59	0.025	0.081
96–125 d-old	0.92 ^b	1.29 ^a	1.16 ^a	0.71 ^c	0.068	<0.001
F/G						
35–65 d-old	3.02	2.93	2.99	3.29	0.160	0.410
66–95 d-old	3.56	4.11	4.21	3.90	0.216	0.160
96–125 d-old	3.32 ^b	4.39 ^a	2.95 ^b	3.49 ^b	0.211	0.002

5. Given the nature of the dietary intervention (probiotics and synbiotics supplementation) and the expectation of major changes in microbiota composition and metabolism, measuring small intestinal microbiota seems critical in this study. Why only colonic microbiota was measured?

Response: The colon is the main site for microbial fermentation, so this study mainly measured colonic microbiota.

6. Which ANOVA was used for statistical analysis? One/two way?

Response: one-way ANOVA was used for statistical analysis. We have added this in the revised manuscript (L254).

7. How about the growth performance of the offspring piglets after dietary supplementation? What about mortality rate during the experimental period from 35 to 125 days?

Response: The growth performance (including BW, ADG, ADFI, and F/G) of offspring pigs have been reported in our previous study (Zhu et al., 2023). No pigs died during the trial.

Zhu Q, Azad MAK, Dong H, Li C, Li R, Cheng Y, Liu Y, Yin Y, Kong X. 2023. Sow-offspring diets supplemented with probiotics and synbiotics are associated with offspring's growth performance and meat quality. *Int J Mol Sci* 24:7668.

8. Line 100 "Sows were housed in individual pens (2.2 × 0.6 m) during gestation"

in the pigging of the trail what is the number of sow month gestation?

Response: Approximately 64 pregnant sows (within a month) were assigned for this trial. The average gestation period of sows was 114 ± 1.58 days.

9. How long did the sows receive feed with experimental supplements? Whole pregnancy or less? It needs to be added.

Response: The sows received feed with experimental supplements during gestation and lactation periods. We have added details in lines 114-115 this in the revised manuscript.

10. Lines 113-114: although the supplier, ingredient, and feeding method of these additives were consistent with the previous study (12), it is worth giving the name of the probiotic and prebiotic here.

Response: We added the information of probiotic and prebiotic in the manuscript. In addition, we also answered this in the first question.

11. What were the criteria for eight piglets for sampling (L118-119)?

Response: Eight offspring pigs per group (one pig from each pen with an average BW of the pen) at each time point (65, 95, and 125 d-old) were selected for sampling. We have added these details in the revised manuscript (L148-149).

12. What about the environmental conditions surrounding the sows and piglets? For example, season, temperature, relative humidity, the nature of housing, and the form of feed provided?

Response: The sows were feeding in the autumn and winter, and the piglets were feeding in the spring and summer. Feeding and housing management were performed according to the standard operations of commercial pig farms.

13. Why is there a difference in the amounts of probiotics, synbiotics, and antibiotics during the different trial periods? If the addition is to the feed, then the difference in the amount of feed consumed between the sow and the offspring is variable, and not the difference in the concentration of the feed additives used.

Response: There were differences in the requirements of sows and piglets (details provided in the "Animals and diets" part). The doses of the probiotics and synbiotics were as recommended by the manufacturers.

14. Are there any data on pregnant sows during the first period of the experiment, such as the rate of feed consumption, average weight of the offspring pigs at birth, as well as their body weight at weaning and mortality rate during lactation period as affected by feed additives used?

Response: Response: The effects of these additives on pregnant and lactating sows, as well as suckling piglets, were reported in our previous studies.

Ma C, Zhang W, Gao Q, Zhu Q, Song M, Ding H, Yin Y, Kong X. 2020. Dietary synbiotic alters plasma biochemical parameters and fecal microbiota and metabolites

in sows. *J Funct Foods*, 75:104221.

Ma C, Gao Q, Zhang W, Zhu Q, Kong X. 2020. Effects of dietary lactobacillus and yeast fermentation broth on reproductive performance colostrum composition and plasma biochemical indexes of sows. *Chinese Journal of Animal Nutrition* 32(1):129–137.

Ma C, Gao Q, Zhang W, Zhu Q, Tang W, Blachier F, Ding H, Kong X. 2020. Supplementing synbiotic in sows' diets modifies beneficially blood parameters and colonic microbiota composition and metabolic activity in suckling piglets. *Front Vet Sci* 7:575685.

Ma C, Azad MAK, Tang W, Zhu Q, Wang W, Gao Q, Kong X. 2022. Maternal probiotics supplementation improves immune and antioxidant function in suckling piglets via modifying gut microbiota. *J Appl Microbiol* 133:515–528.

15 Authors should re-format the references based on journal format See the instructions for authors

Response: We have re-formatted the references based on journal format.

Re: mSystems00048-24R1 (Dietary probiotics and synbiotics supplementation starting from maternal gestation improves muscular lipid metabolism in offspring piglets by reshaping colonic microbiota and metabolites)

Dear Prof. Xiangfeng Kong:

Thank you for the privilege of reviewing your work. Below you will find my comments and instructions from the mSystems editorial office.

Thank you for this revision, the reviewers were satisfied with your answers. Therefore, before accepting your manuscript for publication in mSystems, I need you to update the Data Availability section. Please make the data available as explained in the ASM Data Policy, and as I asked before the revision.

Revision Guidelines

Sincerely,
Juliette Hayer
Editor
mSystems

Response letter

Reviewer #1 (Comments for the Author):

The work entitled "Dietary probiotics and synbiotics supplementation starting from maternal gestation improves muscular lipid metabolism in offspring piglets by reshaping colonic microbiota and metabolites" by Zhu et al. provided valuable data about the application of feed additives (probiotics and synbiotics) subsequently to sows and their offspring piglets on the skeletal muscle lipid metabolism and gut microbiota and metabolites. The authors generated a lot of data to verify their hypothesis, and their findings showed that application of probiotics and synbiotics to sows and offspring is beneficial to improve the quality through modulating gut microbiota and lipid metabolism of offspring pigs. Generally, this manuscript was well-designed and carefully organized. However, there are still some minor concerns worthy of the attention of the authors:

Response: We would like to thank you for your valuable time and insightful comments and suggestions on our manuscript. The comments and suggestions are helpful to improve the quality of our manuscript. We have read all comments and suggestions carefully and made corrections accordingly.

Detailed comments:

1. What amounts of feed were provided daily? Restricted or *ad libitum*?

Response: The sows were fed with 0.8, 1.0, 1.2, 1.5, and 2.0 kg of pregnancy diets from days 1–15, 16–30, 31–75, 76–90, and 91–105 of pregnancy, respectively; fed with 1 kg of pregnancy feed diets a week before parturition and *ad libitum* access after three days of parturition; and fed with 2.4 kg of lactation diets until weaning. The offspring pigs had *ad libitum* to feed at all times.

We have added these details in the revised manuscript (L117-121 and L129-130).

2. What about the sexual distribution of the experimental pigs? Male/female? Provide more details.

Response: One male and one female piglet close to the average BW per litter were selected and transferred to the nursery house for the subsequent feeding trial. We have added this detail in the revised manuscript (L124-125).

3. Was there any vaccination program for sows and offspring piglets during the trial?

Response: Feeding and management (including vaccination program) for sows and offspring piglets were carried out according to the standard operations of commercial pig farms. We have added these in the revised manuscript (L143-415).

4. What were the criteria for eight piglets for sampling (L118-119)?

Response: Eight offspring pigs per group (one pig from each pen with an average BW of the pen) at each time point (65, 95, and 125 day-old) were selected for sampling. We have added these details in the revised manuscript (L148-149).

5. Which ANOVA was used for statistical analysis? One/two way?

Response: The one-way ANOVA was used for statistical analysis. We have added this in the revised manuscript (L254).

6. Keep the consistent format throughout the text ($P < 0.05$ or $P < 0.05$).

Response: We have checked and revised throughout the manuscript carefully.

7. What other health indexes (L577)?

Response: This sentence means that the changes in fatty acid composition could improve the fatty acid content related to the health-promoting index. To avoid ambiguity, we removed "and several indexes related to beneficial health" in the revised manuscript (L607-608).

Minor points:

L6: with these>> of these

Response: We have corrected it (L21).

L16: as well as upregulated...

Response: We have corrected it (L30).

L25-26: different age stages.

Response: Corrected (L40-41).

L110-113: mL/day>> mL/d; keep consistent format throughout the text.

Response: We have checked and revised these throughout the manuscript.

L123: "Sacrifice" is more likely religious word; replace with the word "euthanized".

Response: We have replaced the word "sacrifice" to "euthanized" (L153).

L124: Location and weight of the colon contents?

Response: Approximately 2 g of the colon contents (middle section) were collected into sterile centrifuge tubes. We have added this in the revised manuscript (L154).

L207/208: $P \leq 0.05$; $VIP \geq 1$.

Response: We have corrected it in the revised manuscript (L237/238).

L210: using the

Response: We have added "the" in the revised manuscript (L240).

L237: Define SOA, SOP, and SOS at their first appearance in the text.

Response: We have added the details about SOA, SOP, and SOS at their first appearance in the text.

L257: C18:1n9t content.

Response: We have revised it in the revised manuscript (L289).

L278: Define MyoG at its first appearance in the text.

Response: There is an error in this place, we have checked the results and revised the results and discussion section.

L306: Details of these 88 samples should be provided.

Response: These 88 samples included 32, 31, and 25 samples at 65, 95, and 125 day-old. We have added this information in the revised manuscript (L337-338).

L 308: $P > 0.05$.

Response: We have corrected it (L341).

L411: $P < 0.05$.

Response: Corrected (L443).

L419: $VIP \{ \text{greater than or equal to} \} 2$.

Response: We have revised it in the revised manuscript (L451).

L564: IMF content.

Response: We have added "content " in the revised manuscript (L596).

L612-613:resulted in upregulation of ATGL, HSL, and CPT-1 expressions...

Response: We have revised it in the revised manuscript (L640-641).

L615: Which specific groups had decreased IMF?

Response: The IMF content was decreased in the SOP group at 65 day-old and SOS group at 125 day-old. We have added this in the revised manuscript (L640-641).

Tables: Please provide table note for "-".

Response: We have provided table note for "-" in Tables 1 and 2.

Table 2: C18:3n3 (95 day-old); Why comparison without detection values of other groups?

Response: We have deleted the comparison without detection values.

Response letter

Reviewer #2 (Comments for the Author):

Probiotics and synbiotics, as common feed additives in livestock production, are widely used to enhance the animals' health and performance. The authors present an interesting question in this paper: Can dietary probiotics and synbiotics supplementation improve muscular lipid metabolism in offspring piglets by reshaping colonic microbiota and metabolites? There is a large amount of presented but not seated in any statements or data that would make the work meaningfully applicable to swine production. My detailed assessment as follows:

Response: Sincerest thanks for your valuable time and constructive comments on our manuscript. We have read all comments and suggestions carefully and made corrections accordingly.

1. Please provide the information of the probiotics and synbiotics products used in the present study, e.g., purity, source, composition, etc.

Response: The probiotics mixture was provided by Hunan Lifeng Biotechnology Co., Ltd. (Changsha, China) and contained *Lactobacillus plantarum* B90 (CGMCC1.12934) $\geq 1 \times 10^8$ CFU/mL and *Saccharomyces cerevisiae* P11 (CGMCC2.3854) $\geq 0.2 \times 10^8$ CFU/mL. The XOS ($\geq 35\%$) was provided by Shandong Longlive Biotechnology Co., Ltd. (Shandong, China) and contained xylobiose (55%), xylotriose (25%), xyloetraose (10%), xylopentose (5%), xylohexaose (3%), and xyloheptaose (2%), which met the feed additive of XOS recommended requirements (GB/T23747-2009). The information reported in our previous study (Zhu Q, Song M, Azad MAK, Cheng Y, Liu Y, Liu Y, Blachier F, Yin Y, Kong X. 2022. Probiotics or synbiotics addition to sows' diets alters colonic microbiome composition and metabolome profiles of offspring pigs. *Front Microbiol* 13:934890.). We also added this information in the revised manuscript (L134-140).

2. Please change [12000] to [12,000], use commas when numbers exceed four digits. Please check and correct the full text.

Response: We have changed [12000] to [12,000] and corrected throughout the full text.

3. What amounts of feed were provided daily? Restricted or ad libitum?

Response: The sows were fed with 0.8, 1.0, 1.2, 1.5, and 2.0 kg of pregnancy diets from days 1–15, 16–30, 31–75, 76–90, and 91–105 of pregnancy, respectively; fed with 1 kg of pregnancy diets a week before parturition and *ad libitum* after three days of parturition; and fed with 2.4 kg of lactation diets until weaning. The offspring pigs had *ad libitum* access to feed at all times.

We have added these details in the manuscript (L117-121 and L129-130).

4. Feed intake and body weight must be provided?

Response: The growth performance (including BW, ADG, ADFI, and F/G) of offspring pigs have been reported in our previous study (Zhu Q, Azad MAK, Dong H, Li C, Li R, Cheng Y, Liu Y, Yin Y, Kong X. 2023. Sow-offspring diets supplemented with probiotics and synbiotics are associated with offspring's growth performance and meat quality. *Int J Mol Sci* 24:7668). The specific data is as follows:

Items	C Group	SOA Group	SOP Group	SOS Group	SEM	P-values
BW, kg						
35 d-old	4.97	5.04	4.71	4.78	0.135	0.275
65 d-old	9.37	9.25	9.49	8.96	0.168	0.162
95 d-old	14.05	15.16	13.52	13.61	0.548	0.151
125 d-old	22.67 ^b	27.23 ^a	28.29 ^a	19.28 ^c	0.912	<0.001
ADG, kg/d						
35–65 d-old	0.15	0.15	0.15	0.14	0.006	0.435
66–95 d-old	0.18	0.17	0.16	0.17	0.012	0.693
96–125 d-old	0.27 ^b	0.30 ^b	0.37 ^a	0.18 ^c	0.023	<0.001
ADFI, kg/d						
35–65 d-old	0.41	0.43	0.42	0.44	0.009	0.312
66–95 d-old	0.64	0.65	0.69	0.59	0.025	0.081
96–125 d-old	0.92 ^b	1.29 ^a	1.16 ^a	0.71 ^c	0.068	<0.001
F/G						
35–65 d-old	3.02	2.93	2.99	3.29	0.160	0.410
66–95 d-old	3.56	4.11	4.21	3.90	0.216	0.160
96–125 d-old	3.32 ^b	4.39 ^a	2.95 ^b	3.49 ^b	0.211	0.002

5. Given the nature of the dietary intervention (probiotics and synbiotics supplementation) and the expectation of major changes in microbiota composition and metabolism, measuring small intestinal microbiota seems critical in this study. Why only colonic microbiota was measured?

Response: The colon is the main site for microbial fermentation, so this study mainly measured colonic microbiota.

6. Which ANOVA was used for statistical analysis? One/two way?

Response: one-way ANOVA was used for statistical analysis. We have added this in the revised manuscript (L254).

7. How about the growth performance of the offspring piglets after dietary supplementation? What about mortality rate during the experimental period from 35 to 125 days?

Response: The growth performance (including BW, ADG, ADFI, and F/G) of offspring pigs have been reported in our previous study (Zhu et al., 2023). No pigs died during the trial.

Zhu Q, Azad MAK, Dong H, Li C, Li R, Cheng Y, Liu Y, Yin Y, Kong X. 2023. Sow-offspring diets supplemented with probiotics and synbiotics are associated with offspring's growth performance and meat quality. *Int J Mol Sci* 24:7668.

8. Line 100 "Sows were housed in individual pens (2.2 × 0.6 m) during gestation"

in the pigging of the trail what is the number of sow month gestation?

Response: Approximately 64 pregnant sows (within a month) were assigned for this trial. The average gestation period of sows was 114 ± 1.58 days.

9. How long did the sows receive feed with experimental supplements? Whole pregnancy or less? It needs to be added.

Response: The sows received feed with experimental supplements during gestation and lactation periods. We have added details in lines 114-115 this in the revised manuscript.

10. Lines 113-114: although the supplier, ingredient, and feeding method of these additives were consistent with the previous study (12), it is worth giving the name of the probiotic and prebiotic here.

Response: We added the information of probiotic and prebiotic in the manuscript. In addition, we also answered this in the first question.

11. What were the criteria for eight piglets for sampling (L118-119)?

Response: Eight offspring pigs per group (one pig from each pen with an average BW of the pen) at each time point (65, 95, and 125 d-old) were selected for sampling. We have added these details in the revised manuscript (L148-149).

12. What about the environmental conditions surrounding the sows and piglets? For example, season, temperature, relative humidity, the nature of housing, and the form of feed provided?

Response: The sows were feeding in the autumn and winter, and the piglets were feeding in the spring and summer. Feeding and housing management were performed according to the standard operations of commercial pig farms.

13. Why is there a difference in the amounts of probiotics, synbiotics, and antibiotics during the different trial periods? If the addition is to the feed, then the difference in the amount of feed consumed between the sow and the offspring is variable, and not the difference in the concentration of the feed additives used.

Response: There were differences in the requirements of sows and piglets (details provided in the "Animals and diets" part). The doses of the probiotics and synbiotics were as recommended by the manufacturers.

14. Are there any data on pregnant sows during the first period of the experiment, such as the rate of feed consumption, average weight of the offspring pigs at birth, as well as their body weight at weaning and mortality rate during lactation period as affected by feed additives used?

Response: Response: The effects of these additives on pregnant and lactating sows, as well as suckling piglets, were reported in our previous studies.

Ma C, Zhang W, Gao Q, Zhu Q, Song M, Ding H, Yin Y, Kong X. 2020. Dietary synbiotic alters plasma biochemical parameters and fecal microbiota and metabolites

in sows. *J Funct Foods*, 75:104221.

Ma C, Gao Q, Zhang W, Zhu Q, Kong X. 2020. Effects of dietary lactobacillus and yeast fermentation broth on reproductive performance colostrum composition and plasma biochemical indexes of sows. *Chinese Journal of Animal Nutrition* 32(1):129–137.

Ma C, Gao Q, Zhang W, Zhu Q, Tang W, Blachier F, Ding H, Kong X. 2020. Supplementing synbiotic in sows' diets modifies beneficially blood parameters and colonic microbiota composition and metabolic activity in suckling piglets. *Front Vet Sci* 7:575685.

Ma C, Azad MAK, Tang W, Zhu Q, Wang W, Gao Q, Kong X. 2022. Maternal probiotics supplementation improves immune and antioxidant function in suckling piglets via modifying gut microbiota. *J Appl Microbiol* 133:515–528.

15 Authors should re-format the references based on journal format See the instructions for authors

Response: We have re-formatted the references based on journal format.

Re: mSystems00048-24R2 (Dietary probiotics and synbiotics supplementation starting from maternal gestation improves muscular lipid metabolism in offspring piglets by reshaping colonic microbiota and metabolites)

Dear Prof. Xiangfeng Kong:

Please add the DOI to the data in the appropriate section "Data availability" in the manuscript. It can be kept in the methods as well but it needs to appear in the dedicated section. I still see the "Data availability" paragraph unchanged in the manuscript.

Your manuscript has been accepted, and I am forwarding it to the ASM production staff for publication. Your paper will first be checked to make sure all elements meet the technical requirements. ASM staff will contact you if anything needs to be revised before copyediting and production can begin. Otherwise, you will be notified when your proofs are ready to be viewed.

Cover Image Submissions: If you would like to submit a potential Cover Image, please email a file and a short legend to msystems@asmusa.org. Please note that we can only consider images that (i) the authors created or own and (ii) have not been previously published. By submitting, you agree that the image can be used under the same terms as the published article. Image File requirements: TIF/EPS, 7.5 inches wide by 8.25 inches tall (at least 2,250 pixels wide by 2,475 pixels tall), minimum 300 dpi resolution (600 dpi preferred), RGB, and no figure elements, e.g., arrows or panel labels. The legend should be a short description of the image, 1-2 sentences recommended.

Sincerely,
Juliette Hayer
Editor
mSystems